

# MadNIS – Neural multi-channel importance sampling

Theo Heimel[1], Ramon Winterhalder[2]⋆, Anja Butter[1,3], Joshua Isaacson[4],
Claudius Krause[1], Fabio Maltoni[2,5], Olivier Mattelaer[2] and Tilman Plehn[1]

**1** Institut für Theoretische Physik, Universität Heidelberg, Germany
**2** CP3, Université catholique de Louvain, Louvain-la-Neuve, Belgium
**3** LPNHE, Sorbonne Université, Université Paris Cité, CNRS/IN2P3, Paris, France
**4** Theoretical Physics Division, Fermi National Accelerator Laboratory, Batavia, IL, USA
**5** Dipartimento di Fisica e Astronomia, Universitá di Bologna, Italy

⋆ ramon.winterhalder@uclouvain.be

## Abstract

Theory predictions for the LHC require precise numerical phase-space integration and
generation of unweighted events. We combine machine-learned multi-channel weights
with a normalizing flow for importance sampling, to improve classical methods for nu-
merical integration. We develop an efficient bi-directional setup based on an invertible
network, combining online and buffered training for potentially expensive integrands.
We illustrate our method for the Drell-Yan process with an additional narrow resonance.

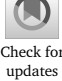

# 1  Introduction

The comparison of data with first-principle predictions defines LHC physics. Event generators provide and evaluate fundamental theory predictions as the key part of a comprehensive forward simulation chain [1]. Given that event generation is an inherently numerical task, it can be improved and accelerated by modern machine learning in, essentially, all aspects [2,3]. In view of the upcoming HL-LHC, such an improvement in speed and precision is crucial to avoid a situation where theory predictions limit the entire relevant LHC program.

Starting with the integration of matrix elements over phase space, we can use neural networks to replace expensive loop amplitudes with fast and precise surrogates [4–8]. The precise knowledge of the amplitude structure can then be used to significantly improve the phase-space integration for a given process [9]. Generally, it is possible to improve numerical integration through neural networks by directly learning the primitive function [10], or using modified and enhanced implementations of importance sampling [11–16]. Technically, this promising approach encodes a change of integration variables in a normalizing flow [17] and then uses *online training* [18] while generating weighted phase space configurations, or weighted events.

This rough online training is successful because normalizing flows, or invertible networks (INNs) [19,20], are especially well-suited, stable, and precise in LHC physics applications [21]. This has been shown in many instances, including event generation [22–25], detector simulations [26–29], unfolding or inverse simulations [20, 30], kinematic reconstruction [31], Bayesian inference [32,33], or inference using the matrix element method [34]. On the other hand, for expensive integrands online training is clearly not optimal, because it does not make use of all previously generated data at subsequent stages of the network training.

For a more efficient training we can use the main structural feature of normalizing flows, their bijective structure best realized in the fully symmetric INN variant introduced in Ref. [19, 35,36]. It allows us to train the same INN online and on previously generated events in parallel. Such a *buffered training* makes optimal use of potentially expensive integrands, but requires a dedicated loss function and training strategy, as we will explain in detail.

In multi-purpose LHC event generators like MADGRAPH5_AMC@NLO [37] (MG5AMC), SHERPA [38] or WHIZARD [39] importance sampling is combined with a multi-channel split of the phase space integration. As it is not guaranteed that an enhanced importance sampling method provides optimal results when combined with standard multi-channel algorithms, we complement our flow-based integration with trainable channel weights. Finally, we introduce a new implementation of rotation layers in the normalizing flow architecture, to aid our ML-importance sampling for high-dimensional phase spaces.

In this paper, we present MADNIS (**Mad**graph-ready **N**eural Networks for Multi-Channel **I**mportance **S**ampling), a comprehensive framework for ML-based phase space sampling ready to be used in a multi-purpose event generator. In Sec. 2, we briefly review the basic concepts of multi-channel integration and importance sampling, before we introduce our new ML-implementations in Sec. 3. We illustrate the ML-channel weights and their interplay with our new bi-directional training for neural importance sampling in Sec. 4. In Sec. 5, we show how our method works for an actual LHC process, the Drell-Yan process with an additional

narrow Z'-resonance. In the Appendix, we provide a detailed description of possible loss functions for our online and buffered training and potential issues with the implementation of this new training approach.

# 2 Classic multi-channel integration

The main structure of LHC phase space generators and integrators is the combination of importance sampling and multi-channel factorization [40]. The reason is that even advanced sampling methods are not powerful enough to probe all phase space features with the required precision, and that we know the leading features from the construction of the helicity amplitudes based on Feynman diagrams. Before we introduce a network-based implementation, we briefly review the standard methods.

## 2.1 Multi-channel decomposition

A generic integral of a function $f \sim |\mathcal{M}|^2$ over the $d$-dimensional phase space $x \in \Phi \subseteq \mathbb{R}^d$ can be represented by

$$I[f] = \int_\Phi d^d x \, f(x). \tag{1}$$

The standard multi-channel method [40, 41], which is also followed by SHERPA, starts by introducing several mappings $G_i : \Phi \to U_i = [0, 1]^d$ denoted as $x \to y = G_i(x)$, of the phase-space variables to obtain individual densities

$$g_i(x) = \left| \frac{\partial G_i(x)}{\partial x} \right|, \qquad \text{with} \qquad \int dx \, g_i(x) = 1, \quad \text{for} \quad i = 1, \dots, m, \tag{2}$$

where $m$ is the total number of channels. Typically, the mappings $G_i(x)$ are initially fixed and based on prior physics knowledge, like the structure of the underlying Feynman diagrams. In practice, current event generators like MG5AMC, SHERPA or WHIZARD do not solely rely on physics-inspired mappings $G_i(x)$, but also combine it with an adaptive VEGAS algorithm [42–46]. Ignoring this for now, the different channels can still be optimized with respect to some channel weights $\alpha_i$ by combining the individual channel densities into a total density

$$g(x) = \sum_i^m \alpha_i g_i(x), \qquad \text{with} \qquad \sum_i^m \alpha_i = 1, \quad \text{and} \quad \alpha_i \geq 0, \tag{3}$$

which also renders $g(x)$ normalized. With this, Eq.(1) becomes

$$I[f] = \sum_i^m \int_\Phi d^d x \, \alpha_i \, g_i(x) \frac{f(x)}{g(x)} = \sum_i^m \int_{U_i} d^d y \, \alpha_i \, \left. \frac{f(x)}{g(x)} \right|_{x = \overline{G}_i(y)}, \tag{4}$$

where $\overline{G}$ denotes the inverse transformation to $G$. The optimization finds the set of global $\alpha_i$ that minimizes the total variance [40, 41].

The single-diagram-enhanced method in MG5AMC [47, 48] defines local, phase-space dependent, channel weights $\alpha_i(x)$ as

$$f(x) = \sum_i^m \alpha_i(x) f(x), \qquad \text{with} \qquad \sum_i^m \alpha_i(x) = 1, \quad \text{and} \quad \alpha_i(x) \geq 0. \tag{5}$$

Inserting this into Eq.(1), we can decompose and parameterize the phase-space integral as

$$I[f] = \sum_i^m \int_\Phi \mathrm{d}^d x \; \alpha_i(x) f(x) = \sum_i^m \int_{U_i} \mathrm{d}^d y \; \alpha_i(x) \frac{f(x)}{g_i(x)}\bigg|_{x=\overline{G}_i(y)} . \tag{6}$$

Once an appropriate decomposition in terms of $\alpha_i(x)$ is found, the channel weights are fixed and not further optimized. The difference between Eq.(4) and Eq.(6) can be understood just as different channel splittings. If we define the local weights as

$$\alpha_i(x) = \alpha_i \frac{g_i(x)}{g(x)}, \tag{7}$$

the two approaches coincide. For more details about the differences of both multi-channel strategies when used in practice, we refer to Ref. [47].

**Single diagram enhancement**

While for a generic integral, finding suitable weights $\alpha_i(x)$ might be unfeasible, MG5AMC introduces two different sets of $\alpha_i(x)$ for phase-space integration. In the first basis [47], we can parameterize the integral as

$$I[|\mathcal{M}|^2] = \sum_i^m \int_\Phi \mathrm{d}^d x \; \alpha_i(x) |\mathcal{M}(x)|^2, \quad \text{with} \quad \alpha_i(x) = \frac{|\mathcal{M}_i(x)|^2}{\sum_j |\mathcal{M}_j(x)|^2}, \tag{8}$$

where $i$ indicates individual Feynman diagrams. This choice of $\alpha_i$ is motivated by the classical limit without interference,

$$I[|\mathcal{M}|^2] = \sum_i^m \int_\Phi \mathrm{d}^d x \; |\mathcal{M}_i(x)|^2 \frac{|\mathcal{M}(x)|^2}{\sum_j |\mathcal{M}_j(x)|^2} \approx \sum_i^m \int_\Phi \mathrm{d}^d x \; |\mathcal{M}_i(x)|^2 \times 1. \tag{9}$$

In this limit each channel is behaving as a squared diagram, its features are easily identifiable, and importance sampling is easy to implement. In general, the number of channels $m$ are completely arbitrary and will often be less than the number of Feynman diagrams $M$, i.e. $m \leq M$.

An alternative choice of channel weights in MG5AMC [48] replaces the $|\mathcal{M}_i|^2$ by the product of all propagator denominators appearing in a given diagram and normalizes them as needed,

$$\alpha_i(x) = \frac{\bar{\alpha}_i(x)}{\sum_j \bar{\alpha}_j(x)}, \quad \text{with} \quad \bar{\alpha}_i(x) = \prod_{k \in \text{prop}} \frac{1}{|p_k(x)^2 - m_k^2 - im_k\Gamma_k|^2} . \tag{10}$$

While this works extremely well for VBF-like or multi-jet processes, this does not seem to be a good choice for W/Z + jets or $t\bar{t}$ + jets production [48].

## 2.2 Monte-Carlo error

To efficiently calculate an integral, we rely on a smart choice for the variable transformation $y = G(x)$ introduced in Eq.(4),

$$I[f] = \int_\Phi \mathrm{d}^d x \, f(x) = \int_U \mathrm{d}^d y \; \frac{f(x)}{g(x)}\bigg|_{x=\overline{G}(y)}, \quad \text{with} \quad g(x) = \left| \frac{\partial G(x)}{\partial x} \right|, \tag{11}$$

which can be any combination of analytic remappings [41], a VEGAS-like numerical remapping [42–46, 49], or a normalizing flow [13–16]. To construct an optimal variable transformation we need a figure of merit for the phase space integration. While the integral is unchanged under the above reparametrization, the variance $\sigma^2$ of the new integrand is given by

$$\sigma^2 \equiv \sigma^2\left[\frac{f}{g}\right] = \int \mathrm{d}^d x \left(\frac{f(x)}{g(x)} - I[f]\right)^2, \tag{12}$$

and becomes minimal for a perfect mapping with $g(x) = f(x)/I[f]$. In practice, we evaluate the Monte Carlo estimate of our integral with discrete sampled points,

$$I[f] = \int_\Phi \mathrm{d}^d x \, g(x) \frac{f(x)}{g(x)} = \left\langle \frac{f(x)}{g(x)} \right\rangle_{x \sim g(x)} \approx \frac{1}{N} \sum_{j=1}^N \frac{f(x_j)}{g(x_j)}\bigg|_{x_j = \overline{G}(y_j)}. \tag{13}$$

In this case the error of the Monte Carlo estimate is itself estimated through the variance defined in Eq.(12) [41],

$$\Delta_N^2 = \frac{\sigma^2}{N} = \frac{1}{N-1}\left[\left\langle \frac{f(x)^2}{g(x)^2} \right\rangle_{x \sim g(x)} - \left\langle \frac{f(x)}{g(x)} \right\rangle^2_{x \sim g(x)}\right]. \tag{14}$$

Note the correction factor $N/(N-1)$ to obtain the unbiased result.

Next, we split the integral into independent channels, as defined in Eq.(6). The Monte Carlo estimate of the integral is given by the sum of the individual estimates

$$I[f] \approx \sum_i \left\langle \alpha_i(x) \frac{f(x)}{g_i(x)} \right\rangle_{x \sim g_i(x)}, \tag{15}$$

where the individual channels are evaluated using $N_i$ points and $\sum_i N_i = N$. The error on the total integral is given by the uncorrelated combination of the channel-wise errors,

$$\Delta_N^2 = \sum_i \Delta_{N_i,i}^2 = \sum_i \frac{\sigma_i^2}{N_i},$$

$$\text{with} \quad \sigma_i^2 = \frac{N_i}{N_i - 1}\left[\left\langle \alpha_i(x)^2 \frac{f(x)^2}{g_i(x)^2} \right\rangle_{x \sim g_i(x)} - \left\langle \alpha_i(x) \frac{f(x)}{g_i(x)} \right\rangle^2_{x \sim g_i(x)}\right]. \tag{16}$$

As known from stratified sampling [49], the optimal number of points per channel, defined by the minimized combined error is a function of the standard deviations $\sigma_i$

$$N_i = N \frac{\sigma_i}{\sum_k \sigma_k}. \tag{17}$$

In practice, the $\sigma_i$ are calculated during training, and the numbers of points $N_i$ used for the numerical integration are subsequently updated.

## 3  MADNIS

While the state-of-the-art event generators work sufficiently well for simple processes, they require significant computing time for complex LHC processes. Consequently, there have been attempts [11–15] to replace VEGAS [42, 43, 46] with a neural network equivalent. We add several new components to improve the precision of the network-based integrator and sampler.

## 3.1 Neural multi-channel weights

First, MADNIS replaces the local multi-channel weights from Sec.(2.1) with trainable channel-weight networks (CWnets),

$$\alpha_i(x) \to \alpha_i(x|\theta). \tag{18}$$

In analogy to classification networks, we encode the normalization of Eq.(5) into the network architecture. Two possible methods are

$$\bar{\alpha}_i(x|\theta) = \frac{\exp \alpha_i(x|\theta)}{\sum_j \exp \alpha_j(x|\theta)} \in [0,1], \qquad \text{or} \qquad \tilde{\alpha}_i(x|\theta) = \frac{\alpha_i(x|\theta)}{\sum_j \alpha_j(x|\theta)} \in \mathbb{R}. \tag{19}$$

Note that the second normalization also allows for negative channel weights for a generic and unconstrained network output $\alpha_i(x|\theta)$. While this is mathematically allowed and satisfies the requirements in Eq. (2), these channel weights lose their interpretation as probabilities. Our tests, however, indicate that the first version, corresponding to a softmax activation, is more stable during training. We can improve the training by using physics knowledge. For instance, we can learn a correction to a prior weight $\alpha_i^*$ given by MG5AMC,

$$\alpha_i(x|\theta) = \log \alpha_i^*(x) + \theta_i \cdot \Delta_i(x|\theta). \tag{20}$$

This specific form gives the normalized weight

$$\bar{\alpha}_i(x|\theta) = \frac{\alpha_i^*(x) \cdot \exp\left[\theta_i \cdot \Delta_i(x|\theta)\right]}{\sum_j \alpha_j^*(x) \cdot \exp\left[\theta_j \cdot \Delta_j(x|\theta)\right]}, \qquad \text{with} \qquad \sum_i \alpha_i^*(x) = 1. \tag{21}$$

In addition, we can provide the neural network with derived quantities such as invariant masses alongside the event representation $x$.

## 3.2 Neural importance sampling

Second, MADNIS augments the physics-inspired phase space mappings with an INN [19]

$$y = G_i(x) \to G_i(x|\varphi), \qquad \text{and} \qquad x = \overline{G}_i(y|\varphi). \tag{22}$$

This replaces the classic importance sampling density $g_i(x)$ with a network-based variable transformation $g_i(x|\varphi)$ in Eqs.(6) and (15)

$$I[f] = \sum_i \int_{U_i} d^d y \; \alpha_i(x) \frac{f(x)}{g_i(x|\varphi)}\bigg|_{x=\overline{G}_i(y|\varphi)}, \quad \text{with} \quad g_i(x|\varphi) = \left|\frac{\partial G_i(x|\varphi)}{\partial x}\right|, \tag{23}$$

where we assume the latent distribution in $y$ to be uniform. The INN-encoded phase space mapping is trained to provide a surrogate density

$$g_i(x|\varphi) \approx f_i(x) = \alpha_i(x)f(x). \tag{24}$$

The INN variant of a normalizing flow, illustrated in Fig. 1, ensures that the training and the evaluation of the network are symmetric and equally fast in both directions. We will make use of this structural advantage in our training setup.

To clearly separate the discussion of the neural importance sampling from the channel weights defined in Eq.(18), $\alpha_i(x|\theta)$, we denote its network weights as $\varphi$. In principle, the bijective mapping $G_i(x|\varphi)$ can be any combination of a fixed physics-inspired mapping and a normalizing flow.

Normalizing flows are already used to improve numerical integration over phase space [14] or the Feynman parameters in loop integrations [16]. The standard i-flow algorithm [13,15] for importance sampling is

1. Draw samples from the latent space $y \sim$ uniform;

2. Transform them into phase-space points $x = \overline{G}(y|\varphi)$, without gradient calculation;

3. Evaluate the integrand or target distribution $f(x)$;

4. Pass the network in the other direction, $y = G(x|\varphi)$, to evaluate the density $g(x|\varphi)$;

5. Compute divergence-based loss between $f(x)$ and $g(x|\varphi)$;

6. Compute gradients of the loss and optimize the network.

We illustrate the algorithm in Fig. 2. The additional pass in step 4 is important to evaluate $g(x|\varphi)$ as a proper function of $x$ and obtain the correct gradients for training, as explained in the Appendix. Note that the two passes in step 2 and 4 are inverse to each other. We refer to this approach as online training, because the training data $x$ is continuously generated and immediately used once for training. It implies that a potentially expensive integrand $f(x)$ has to be evaluated for every event used to train the network, which makes it inefficient. One way to alleviate this problem is to buffer already generated samples and use them for a limited number of training passes [18].

### 3.3 Buffered training

An alternative training method for the phase-space mapping would be traditional sample-based training, where the same samples can be used every epoch. Pure sample-based training only requires one pass through the INN, but it is not a sensible choice for neural importance sampling, because all training data needs to be available from the beginning. Instead, we iterate between online training, where samples are generated and directly used for training, and buffered training on previously generated events. Because memory constraints inhibit storing all generated phase-space points, we only save a fraction of events in a buffer which is replaced during the next online training phase.

Before looking into the training algorithm in detail, we need to define a common loss function for online and the buffered training, so the combination converges towards a common minimum. The buffered loss has to account for the fact that training happens after sampling, so the network weights will change in between. The sampling probability $q_i(x|\hat{\varphi})$ is different from the density $g_i(x|\varphi)$ at the time of training, even though the two might be related as

$$g_i(x|\varphi) \xrightarrow{\varphi \to \hat{\varphi}} q_i(x|\hat{\varphi}). \tag{25}$$

Consequently, the buffered form of a KL-loss has to be modified according to

$$\mathcal{L} \to \mathcal{L} \times \frac{g_i(x|\varphi)}{q_i(x|\hat{\varphi})}, \tag{26}$$

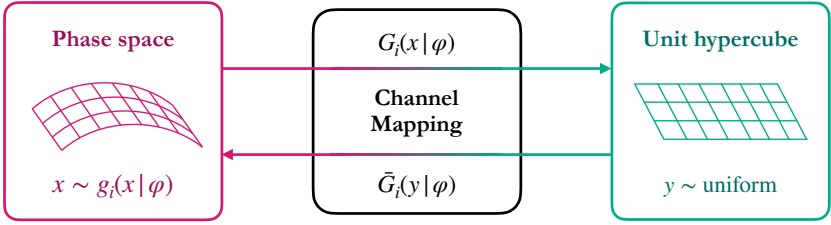

Figure 1: Structure of the INN channel mappings. The latent space $y \sim$ uniform is mapped onto the phase space $x \sim g_i(x|\varphi)$ for each channel $i$.

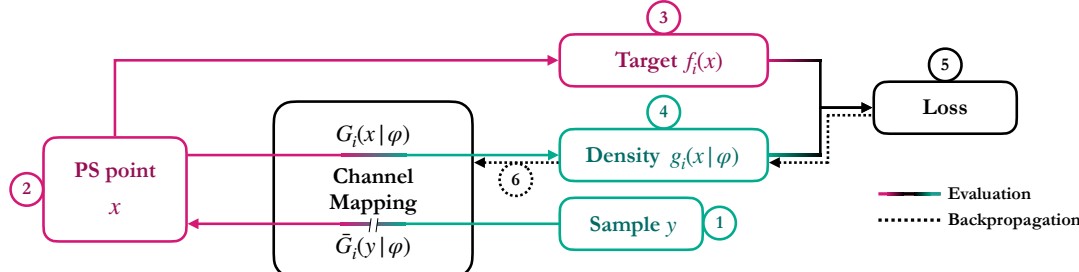

Figure 2: Workflow of the online training of the INN. The discontinuous line from (1) to (2) indicates that it only allows forward sampling but no gradient backpropagation.

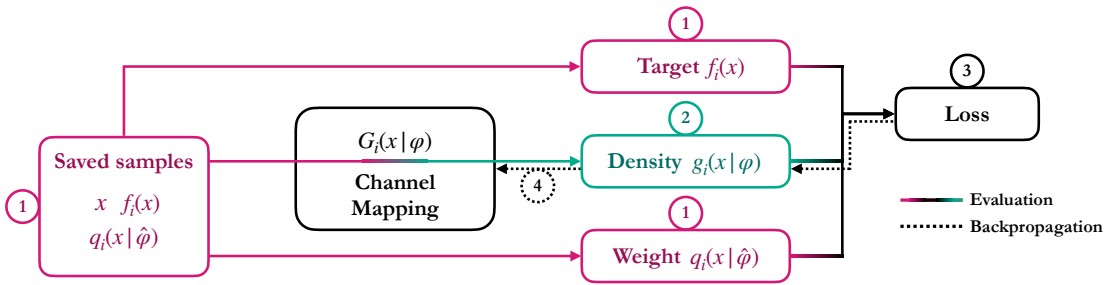

Figure 3: Workflow of the buffered training of the INN.

which is a generalization of the weighted log-likelihood loss in Ref. [22]. This means we have to buffer $x$, $f_i(x)$, and the sampling density $q_i(x|\hat{\varphi})$ to be able to evaluate the loss. More details about the corresponding losses can be found in the Appendix.

In Fig. 3, we illustrate the workflow of the buffered based training:

1. Start with a buffered phase space point $x$ with $f(x)$, and $q_i(x|\hat{\varphi})$.

2. Pass it through the INN and compute the density $g_i(x|\varphi)$.

3. Compute the weighted loss from $g_i(x|\varphi)$ and $f_i(x)$, using $q_i(x|\hat{\varphi})$.

4. Compute gradients and optimize the network.

This training can be combined with the online training introduced in Sec. 3.2, and the balance of the two training strategies can be adjusted depending on how computationally expensive the integrand evaluation is.

**Training time statistics**

To illustrate the trade-off between training time and weight updates, we consider a training taking the time $T$, split into buffered ($T_{\text{buff}} = T \cdot (1 - r_@)$) and online ($T_@ = T \cdot r_@$) training. Let $t_{\text{buff}}$ and $t_@$ be the time for a weight update in the buffered training and online training (excluding the integrand evaluation), respectively. Note that $t_@$ requires an additional sampling without gradient updates, as explained in the Appendix. We find that $t_@/t_{\text{buff}} \approx 1.33$.

If $t_f$ is the time it takes to evaluate the integrand, the time for a weight update in online training will be $t_@ + t_f$, compared to $t_{\text{buff}}$ for the buffered training. The number of weight updates is divided between the training modes,

$$n = n_{\text{buff}} + n_@ = \frac{T(1 - r_@)}{t_{\text{buff}}} + \frac{Tr_@}{t_@ + t_f} \,. \tag{27}$$

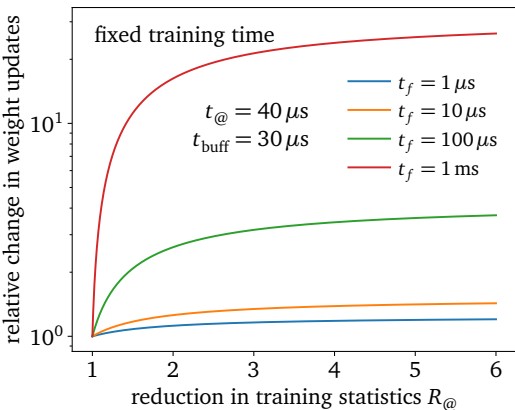
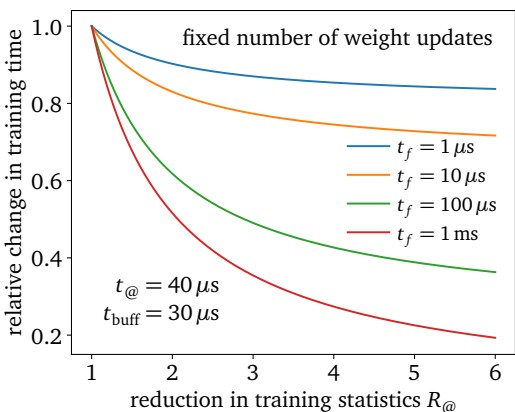

Figure 4: Hypothetical change in weight updates (left panel) and training time (right panel) as a function of the reduction in training statistics $R_@$ for integrands with different computational costs.

As a baseline we can look at the number of weight updates $n_{\text{base}} = T/(t_@ + t_f)$ for pure online training, giving a increase factor in weight updates of

$$\frac{n}{n_{\text{base}}} = \left(1 - \frac{1}{R_@}\right)\frac{t_@ + t_f}{t_{\text{buff}}} + \frac{1}{R_@}, \qquad \text{with} \qquad R_@ = \frac{n_{\text{base}}}{n_@} = \frac{1}{r_@}, \qquad (28)$$

in terms of the reduction factor in training statistics $R_@$ which coincides with the inverse of the relative training time $r_@$. The left panel of Fig. 4 shows the increase factor in weight updates for integrands with different computational cost $t_f$; $t_@$ and $t_{\text{buff}}$ are extracted from a test run on a CPU. In a similar fashion, we can also fix the number of weight updates $n$ and instead compare the reduction factor in training time $T/T_{\text{base}}$ depending on $R_@$, which is shown in the right panel of Fig. 4.

**Variance-weighted training**

Stratified sampling [49] minimize the variances discussed in Sec. 2.2, but it can also improve the network training. We use the variance from Eq.(17) to sample more events in poor channels and weight them accordingly in the loss function. This forces the network to focus on improving these channels, which should ultimately lead to a better convergence of the network. Such a variance-weighted training can be easily combined with both, online and buffered training. To stabilize the training when using variance-weighted channel sampling, we fix a small fraction of events to be uniformly distributed across all channels. This guarantees that no channel is empty during training, which would otherwise lead to an error. This is in contrast to integration and pure sampling, where the algorithm is encouraged to ignore channels with vanishing contributions.

## 3.4 Trainable rotations

The INN employed in our study is based on a bipartite architecture [35, 36] and requires permutations in the order of the coordinates between the coupling blocks to learn all correlations. The simplest implementation is an exchange of the bipartite sets [35, 36]. It ensures that correlations between the variables can be learned stacking a few coupling blocks. Shuffling the elements of the two sets with each other is more efficient, but comes with a small probability that some elements are never modified. Another solution is a deterministic set of permutations based on a logarithmic decomposition of the integral dimension [13]. It ensures that every

pair of elements appears in different bipartite sets at least once. This relates the number of required coupling layers to the dimensionality of the integrand, and is particularly efficient for integrals of dimension $d = 2^k$.

For an integration over $\mathbb{R}^d$ we can generalize these permutations to rotations described by $SO(d)$. Introduced in the context of image generation, a randomly initialized but fixed $SO(d)$ matrix (soft permutation [20]) allows for mixing of color channel information [20,50]. A trainable implementation [50] first adjusts all $d^2$ parameters and then projects the trained matrix back onto $SO(d)$. This implementation as a independent $d \times d$ matrix with a subsequent projection is not efficient.

**Generalized Euler angles**

We construct a trainable soft permutation that only optimizes the relevant degrees of freedom. The elements of $SO(d)$ are described by a $d(d-1)/2$-dimensional Lie algebra and can be parametrized by $D = d(d-1)/2$ real parameters, interpreted as angles. The common parametrization of rotations in $\mathbb{R}^3$ are the Euler angles [51]. They can be generalized to $\mathbb{R}^d$ [52]. To efficiently construct our rotation matrix $R$, we start with an orthonormal basis $\vec{a}_i$, connected to the standard basis $\vec{e}_i$ by

$$\vec{a}_k = \sum_{i=1}^{d} \vec{e}_i R_{ik} \quad \longleftrightarrow \quad \vec{e}_i = \sum_{i=1}^{d} R_{ki} \vec{a}_i. \tag{29}$$

To properly construct the corresponding rotation matrix we proceed iteratively:

1. Define one direction with the unit-vector $\vec{a}_d$ in terms of $d-1$ angles $\vartheta_i^{(d)}$;

2. Construct an orthonormal basis $\{\vec{b}_i^{(d)}\}$, which contains $\vec{a}_d$ as last basis vector;

3. Fix next direction $\vec{a}_{d-1}$ in terms of $d-2$ angles $\vartheta_i^{(d-1)}$ and construct new basis $\{\vec{b}_i^{(d-1)}\}$;

4. Iterate until the basis $\{\vec{a}_i\}$ determines $R$.

For the three steps of this algorithms we provide the details below.

**1. Definition of unit-vector $\vec{a}_d$**   We start by defining the unit-vector $\vec{a}_d$ in terms of $d-1$ angles $\vartheta_i^{(d)}$ or $d$-dimensional spherical coordinates,

$$
\begin{aligned}
\vec{a}_d = {} & \sin\vartheta_1^{(d)}\vec{e}_1 \\
& + \cos\vartheta_1^{(d)}\sin\vartheta_2^{(d)}\vec{e}_2 \\
& + \quad \vdots \qquad\qquad \ddots \\
& + \cos\vartheta_1^{(d)}\dots\cos\vartheta_{d-2}^{(d)}\sin\vartheta_{d-1}^{(d)}\vec{e}_{d-1} \\
& + \cos\vartheta_1^{(d)}\dots\cos\vartheta_{d-2}^{(d)}\cos\vartheta_{d-1}^{(d)}\vec{e}_d = \sum_{i=1}^{d}\vec{e}_i \frac{\sin\vartheta_i^{(d)}}{\cos\vartheta_i^{(d)}}\prod_{j=1}^{i}\cos\vartheta_j^{(d)}, \qquad \text{with} \quad \sin\vartheta_d^{(d)}=1.
\end{aligned}
\tag{30}
$$

While $\cos\vartheta_k^{(d)}$ are assumed to be positive, $\cos\vartheta_{d-1}^{(d)}$ can have either sign.

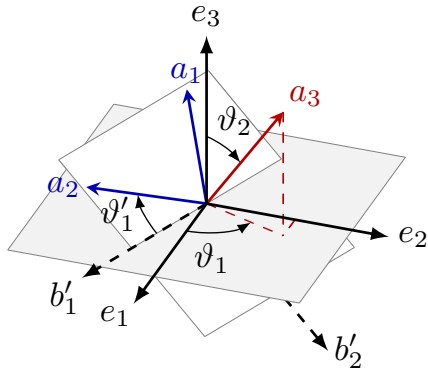

Figure 5: Exemplary rotation of the standard 3-dimensional basis $\vec{e}_i$ into another orthonormal basis $\vec{a}_i$ parametrized by three Euler angles $\vartheta_i$. The different colors of the new basis $\vec{a}_i$ indicate there iterative construction.

**2. Orthonormal basis $\vec{b}_i^{(d)}$** To construct an orthonormal basis which contains $\vec{a}_d$ as one of its basis vectors, we can define the new basis

$$\vec{b}_k^{(d)} = \left(\prod_{j=1}^{k-1}\cos\vartheta_j^{(d)}\right)^{-1}\frac{\partial\vec{a}_d}{\partial\vartheta_k^{(d)}} \qquad (k=1,...,d-1), \qquad \text{and} \qquad \vec{b}_d^{(d)} = \vec{a}_d. \tag{31}$$

This definition fulfills the orthogonality and normalization condition [52], $\vec{b}_i^{(d)}\cdot\vec{b}_k^{(d)} = \delta_{ik}$. The rotation into this basis is given by

$$\vec{b}_k^{(d)} = \sum_i \vec{e}_i A_{ik}^{(d)}, \qquad \text{with} \qquad A^{(d)} = \begin{pmatrix} & & \text{IV} & \\ & \text{I} & & \text{II} \\ \text{III} & & & \end{pmatrix}. \tag{32}$$

In the regions we have

$$\text{I} \qquad A_{ii}^{(d)} = \cos\vartheta_i^{(d)}, \qquad\qquad\qquad \text{for} \quad i=1\ldots d-1,$$

$$\text{II} \qquad A_{id}^{(d)} = \frac{\sin\vartheta_i^{(d)}}{\cos\vartheta_i^{(d)}}\prod_{j=1}^{i}\cos\vartheta_j^{(d)}, \qquad\qquad \text{for} \quad i=1\ldots d,$$

$$\text{III} \qquad A_{ik}^{(d)} = -\frac{\sin\vartheta_i^{(d)}\sin\vartheta_k^{(d)}}{\cos\vartheta_i^{(d)}\cos\vartheta_k^{(d)}}\prod_{j=k}^{i}\cos\vartheta_j^{(d)}, \qquad \text{for} \quad i>k,$$

$$\text{IV} \qquad A_{ik}^{(d)} = 0, \qquad\qquad\qquad\qquad\qquad \text{for} \quad i<k<d. \tag{33}$$

For $\vartheta_1^{(d)} = \cdots = \vartheta_{d-1}^{(d)} = 0$ this gives $A_{ik}^{(d)} = \delta_{ik}^{(d)}$, so the transformation is continuously connected to the identity and $\det A^{(d)} = 1$.

**3. Subsequent basis vectors $\vec{b}_i^{(l)}$** Next, we consider $\vec{a}_{d-1}$ in $\{\vec{b}_1^{(d)},\ldots,\vec{b}_{d-1}^{(d)}\}$. As in Eq.(30) we define this vector in terms of a new set of $d-2$ angles $\vartheta_i^{(d-1)}$ and construct an orthonormal basis $b_i^{(d-1)}$ which contains $\vec{a}_{d-1}$. Similarly, we can proceed for the remaining vectors $\vec{a}_{d-2},\ldots,\vec{a}_2$. A general step $l$ in this iterative basis transformation leads from a basis

$$\vec{b}_1^{(l+1)},\ldots,\vec{b}_l^{(l+1)}, \quad \text{and} \quad \vec{b}_{l+1}^{(l+1)} = \vec{a}_{l+1},\ldots,\vec{b}_d^{(l+1)} = \vec{a}_d, \tag{34}$$

to the basis

$$\vec{b}_1^{(l)}, \dots, \vec{b}_{l-1}^{(l)}, \quad \text{and} \quad \vec{b}_l^{(l)} = \vec{a}_l, \dots, \vec{b}_d^{(l)} = \vec{a}_d, \tag{35}$$

where we have defined $\vec{a}_l$ by

$$\vec{a}_l = \sum_{i=1}^{l} \vec{b}_i^{(l+1)} \frac{\sin \vartheta_i^{(l)}}{\cos \vartheta_i^{(l)}} \prod_{j=1}^{i} \cos \vartheta_j^{(l)}, \qquad \text{with} \quad \sin \vartheta_l^{(l)} = 1. \tag{36}$$

The corresponding transformation into this basis is defined by

$$\vec{b}_k^{(l)} = \sum_i \vec{b}_i^{(l+1)} B_{ik}^{(l)}, \tag{37}$$

where $B^{(l)}$ is the matrix

$$B^{(l)} = \left( \begin{array}{c|c} A^{(l)} & 0 \\ \hline 0 & \mathbb{1}^{(d-l)} \end{array} \right), \tag{38}$$

and where $A^{(l)}$ is defined following Eq.(32)–(33) with angles $\vartheta_i^{(l)}$.

**4. Iteration**  At the end of the procedure, we have an orthonormal basis defined by $d(d-1)/2$ angles which yields the desired $d$-dimensional rotation matrix

$$R = B^{(d)} B^{(d-1)} \dots B^{(3)} B^{(2)}, \tag{39}$$

as introduced in Eq. (29). An illustration of this procedure in 3 dimensions is shown in Fig. 5: First the new basis vector $\vec{a}_3$ is defined by rotations with angles $\vartheta_1$ and $\vartheta_2$. Using Eq.(31) we can construct the new basis $\vec{b}_1', \vec{b}_2', \vec{a}_3$. Afterwards, we define the vector $\vec{a}_2$ in this basis by a rotation with angle $\vartheta_1'$ which also fixes the last basis vector $\vec{b}_1'' = \vec{a}_1$ and determines the procedure. We implement these angles $\vartheta_i^l$ as trainable parameters.

## 4  Toy examples

To check and benchmark the various ideas presented in Sec. 3 we first consider two parametric toy models, a 1-dimensional camel back, and a 2-dimensional crossed ring. The camel back allows us to illustrate how to train channel weights to optimize a simple bi-modal integration. The crossed ring we use to illustrate how learnable local channel weights can be combined with an INN-importance sampling successfully. A discussion of the trainable rotations and the mixed online and buffered training will only become relevant for the LHC example in Sec. 5.

### 4.1  One-dimensional camel back

Our first toy example just illustrates how the neural integrator learns channel weights for pre-defined channels. We define a normalized 1-dimensional camel back or Gaussian mixture,

$$f_{\text{GM}}(x) = \frac{a_1}{\sqrt{2\pi}\sigma_1} \exp\left[-\frac{(x-\mu_1)^2}{2\sigma_1^2}\right] + \frac{1-a_1}{\sqrt{2\pi}\sigma_2} \exp\left[-\frac{(x-\mu_2)^2}{2\sigma_2^2}\right],$$
$$\text{with} \qquad \mu_1 = 2, \qquad \sigma_1 = 0.5, \qquad \mu_2 = 5, \qquad \sigma_2 = 0.1, \qquad a_1 = 0.35. \tag{40}$$

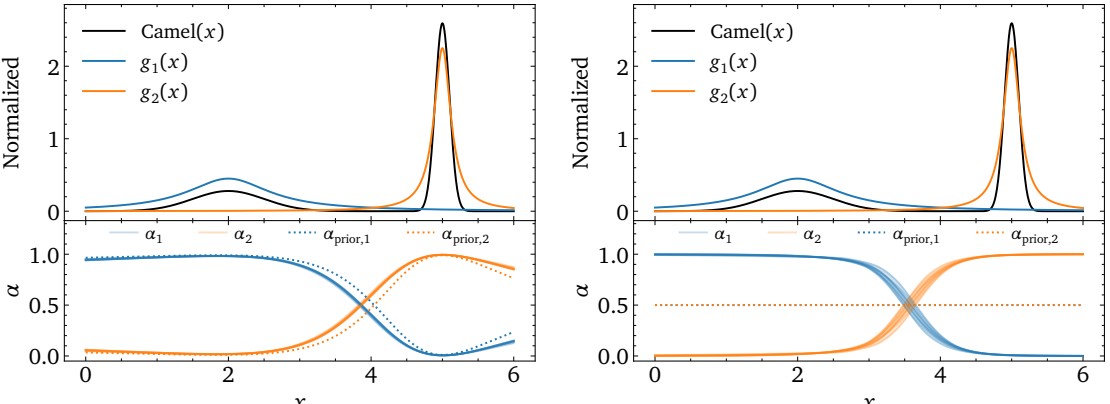

Figure 6: Learned weights for the camel back function for ten different trainings. We train NN-weights starting from a near-optimal (left) or flat (right) prior. The prior weights are illustrated as dotted lines.

If we want to describe each of the hardly overlapping Gaussians by an integration channel we need reasonable mappings which should not be identical to the Gaussian integrand. We choose a Cauchy or Breit-Wigner mapping [53]

$$x = \overline{G}_i(y) = \mu_i + \sqrt{2}\sigma_i \tan\left[\pi\left(y - \frac{1}{2}\right)\right],$$

$$g_i(x) = \frac{1}{\pi} \frac{\sqrt{2}\sigma_i}{(x - \mu_i)^2 + 2\sigma_i^2}. \tag{41}$$

With these definitions the widths of the Gaussian and the Breit-Wigner functions are roughly the same. The multi-channel form of Eq.(6) using a known mapping is

$$I[f_{\text{GM}}] = \int_{-\infty}^{\infty} dx \, f_{\text{GM}}(x) = \sum_{i=1}^{2} \int_{-\infty}^{\infty} dx \, \alpha_i(x|\theta) f_{\text{GM}}(x)$$

$$= \sum_{i=1}^{2} \int_{0}^{1} dy \, \alpha_i(x|\theta) \left. \frac{f_{\text{GM}}(x)}{g_i(x)} \right|_{x = \overline{G}_i(y)}. \tag{42}$$

As mentioned above, the camel back toy model only serves as an illustration that a simple regression network can learn the channel weights $\alpha_i(x|\theta)$, as described in Sec. 3.1. We provide the hyperparameters for this simple network to the left in Tab. 1. The only noteworthy setting is that the loss function of the network is defined as the variance of the integral given in Eq.(16). The amount of training data is comparably large, to give the network a chance to learn the channel weights with enough precision and to allow for a test of the stability using an ensemble of networks.

To the right in Tab. 1 we compare the error on the integral just using uniform, constant weights $\alpha_i$, the (nearly) optimal choice $\alpha_i(x) = g_i(x)/\sum_i g_i(x)$, and local channel weights $\alpha_i(x|\theta)$ optimizing the actual variance. We see that the optimal and the trained weights provide the same results, significantly improving over the naive choice.

In Fig. 6 we show the target function from Eq.(40), the two pre-defined channels $g_i(x)$, and, in the lower panel, the learned channel weights $\alpha_i(x|\theta)$ and their prior or starting points. For the left and right panels network training starts from the near optimal $\alpha_i(x) \propto g_i(x)$ or a flat prior $\alpha_i(x) = \text{const}$. While the first version converges on the same network weights for ten different trainings, the harder task leads to a small variation in the training outcome. Nevertheless, the two learned channel weights are essentially identical, with the exception of

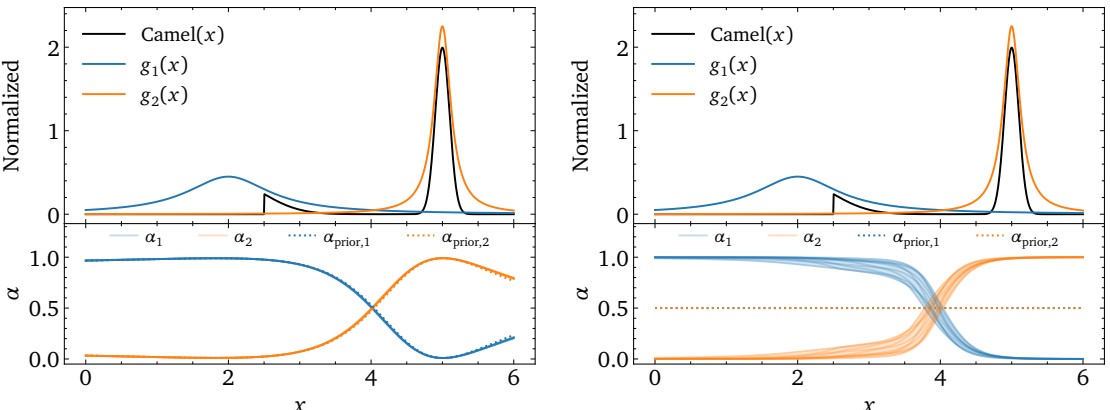

Figure 7: Learned weights for the cut camel back function for ten different trainings. We train NN-weights starting from a near-optimal (left) or flat (right) prior. The prior weights are illustrated as dotted lines.

slight deviations in the exponentially suppressed tails of the two Gaussians. From Tab. 1 we know that these deviations do not have any impact on the evaluation of the integral.

**Camel back with cut**

For the camel-back function in Eq.(40) our well-suited choice of channels $g_i(x)$ in Eq.(41) guarantees that the learned channel weights converge to a reasonable and stable solution. An obvious question is what happens with the trained weights $\alpha_i(x|\theta)$ if the channels $g_i(x)$ are not perfect. To investigate the effect of a non-perfect shape of the channels on the integration we consider a camel back with a cut in the left Gaussian of Eq.(40),

$$f_{\text{GM}}(x) \rightarrow \begin{cases} f_{\text{GM}}(x), & x \geq \mu_1 + \sigma_1, \\ 0, & x < \mu_1 + \sigma_1, \end{cases} \tag{43}$$

where $\mu_1 + \sigma_1 = 2.5$. In Tab. 1 we see that for all methods the integration becomes slightly harder and less numerically reliable. The level of improvement for the network weights remains the same as for the perfect camel back, confirming the power of our NN-channel weights.

Table 1: Left: hyperparameters of the multi-channel weight network for the 1-dimensional camel back. Right: relative errors of the camel back integrals using the trained channel weights (means and standard deviations from ten runs).

| Parameter | Value |
|---|---|
| Loss function | variance |
| Learning rate | 0.001 |
| LR schedule | inverse time decay |
| Decay rate | 0.01 |
| Batch size | 128 |
| Epochs | 20 |
| Batches per Epoch | 100 |
| Number of layers | 3 |
| Hidden nodes | 16 |
| Activation function | leaky ReLU |

| Function | $\alpha_i(x)$ | Rel. Error [%] |
|---|---|---|
| Camel back | Uniform | $2.553 \pm 0.017$ |
| | Optimal | $0.769 \pm 0.006$ |
| | NN (flat prior) | $0.770 \pm 0.005$ |
| | NN (opt. prior) | $0.767 \pm 0.006$ |
| Cut camel back | Uniform | $3.412 \pm 0.048$ |
| | Optimal | $1.031 \pm 0.006$ |
| | NN (flat prior) | $1.032 \pm 0.017$ |
| | NN (opt. prior) | $1.030 \pm 0.009$ |

Based on $10^4$ events

Finally, in Fig. 7 we also see that the modification of the integrand does not affect a properly initialized training, but leads to a slightly larger spread when we train the network from scratch. Such a behavior is expected for any complication of the network task.

## 4.2 Two-dimensional crossed ring

To show how the trained channel weights from Sec. 3.1 and the neural importance sampling from Sec. 3.3 work in combination, we choose a moderately challenging 2-dimensional toy model. It combines a closed Gaussian ring and a diagonal Gaussian line

$$f_{\text{no-parking}}(x) = \frac{1}{2}\left[f_{\text{ring}}(x) + f_{\text{line}}(x)\right],$$

$$f_{\text{line}}(x) = N_1 \exp\left[-\frac{(\tilde{x}_1 - \mu_1)^2}{2\sigma_1^2}\right]\exp\left[-\frac{(\tilde{x}_2 - \mu_2)^2}{2\sigma_2^2}\right],$$

$$f_{\text{ring}}(x) = N_2 \exp\left[-\frac{\left(\sqrt{x_1^2 + x_2^2} - r_0\right)^2}{2\sigma_0^2}\right],$$

$$\text{with} \quad r_0 = 1, \quad \sigma_0 = 0.05, \quad \mu_1 = 0, \quad \sigma_1 = 3, \quad \mu_2 = 0, \quad \sigma_2 = 0.05, \tag{44}$$

where $N_0$ and $N_1$ are chosen such that $f_{\text{ring}}(x)$ and $f_{\text{line}}$ are both normalized to unity and $\tilde{x}_{1,2} = (x_1 \mp x_2)/\sqrt{2}$.

**Channel-mappings**

To see how much additional analytic mappings help, we construct two channels for the line and the ring contributing to our integral. We start with the mapping for the Gaussian line which

Table 2: Left: hyperparameters of the INN and the channel weight network (CWnet) for the crossed ring. The numbers in parentheses indicate that a different setting was used for a ring mapping. Right: Relative integration errors for different numbers of channels and variations of analytic mappings. We show the means and standard deviations for ten independent trainings.

| Parameter | Value |
|---|---|
| Loss function | variance |
| Learning rate | 0.0005 (0.001) |
| LR schedule | inverse time decay |
| Decay rate | 0.02 |
| Batch size | 1024 |
| Epochs | 100 |
| Batches per Epoch | 500 |
| Coupling blocks | affine |
| Permutations | soft |
| Blocks | 6 |
| Subnet hidden nodes | 32 (16) |
| Subnet layers | 3 (2) |
| CWnet layers | 2 |
| CWnet hidden nodes | 16 |
| Activation function | leaky ReLU |

| Fig. | Analytic Mappings | Rel. Error [%] |
|---|---|---|
| 8 | flat | $1.17 \pm 0.13$ |
| 8 | flat, flat | $0.71 \pm 0.15$ |
| 8 | flat, flat, flat | $0.50 \pm 0.15$ |
| 9 | ring, flat | $0.30 \pm 0.11$ |
| | ring, line | $0.14 \pm 0.06$ |
| | ring, line, flat | $0.29 \pm 0.14$ |

Based on $10^4$ events

first aligns the line with the $x_1$-axis by performing a first change of variables $x \to y = G_1(x)$ as

$$x_{1,2} = \frac{y_2 \pm y_1}{\sqrt{2}}, \qquad \text{with} \qquad g_1(x) = \left| \frac{\partial G_1(x)}{\partial x} \right| = 1. \qquad (45)$$

As for the camel back, Eq.(41), we approximate the Gaussian peak through a Breit-Wigner distribution using the variable transformation $y \to z = G_2(y)$,

$$y_{1,2} = \mu_{1,2} + \gamma_{1,2} \tan\left[\pi\left(z_{1,2} - \frac{1}{2}\right)\right], \qquad \text{with} \qquad g_2(y) = \frac{1}{\pi^2} \prod_{j=1}^{2} \frac{\gamma_j}{\gamma_j^2 + (y_j - \mu_j)^2}. \qquad (46)$$

The combined channel density is then $g_{\text{line}}(x) = 1 \times g_2(G_1(x))$. The Gaussian ring requires a mapping $x \to (r, \theta) = G_3(x)$ into polar coordinates

$$x_1 = r \cos\theta, \qquad \text{and} \qquad x_2 = r \sin\theta. \qquad (47)$$

Its Jacobian is $g_3(x) = r$. Again, we approximate the radial peak by a Breit-Wigner through the variable transformation $(r, \theta) \to z = G_4(r, \theta)$,

$$r = r_0 + \gamma_0 \tan\left[\pi(\omega_0 z_1 - C_0)\right],$$

$$\theta = 2\pi z_2, \qquad \text{with} \qquad g_4(r) = \frac{1}{2\pi} \frac{1}{\omega_0 \pi} \frac{\gamma_0}{\gamma_0^2 + (r - r_0)^2}, \qquad (48)$$

where $\pi C_0 = \arctan(r_0/\gamma_0)$ and $\omega_0 = (1 + 2C_0)/2$ ensures $r > 0$ and thus $g_{\text{ring}}(x) = r \, g_4(G_3(x))$. We either augment or replace these mappings with a neural channel mapping $G_i(x|\varphi)$. To challenge our INN when paired with the above mappings we pick wide channel widths,

$$\gamma_{0,1,2} = \sqrt{40} \, \sigma_{0,1,2}. \qquad (49)$$

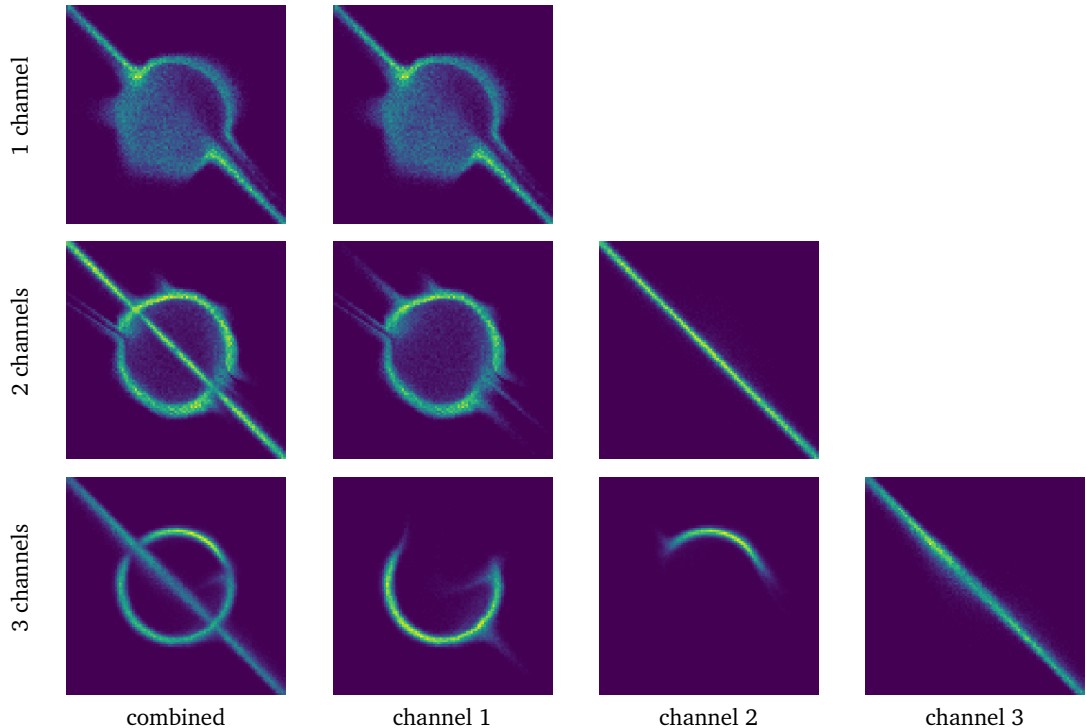

Figure 8: Combined and channel-wise (the latter not weighted by channel weights) distributions learned by a one-, two- and three-channel integrator with flat mappings and a mode-specific prior. Note that the splitting in the three-channel case is not unique and learned differently by the network for each run.

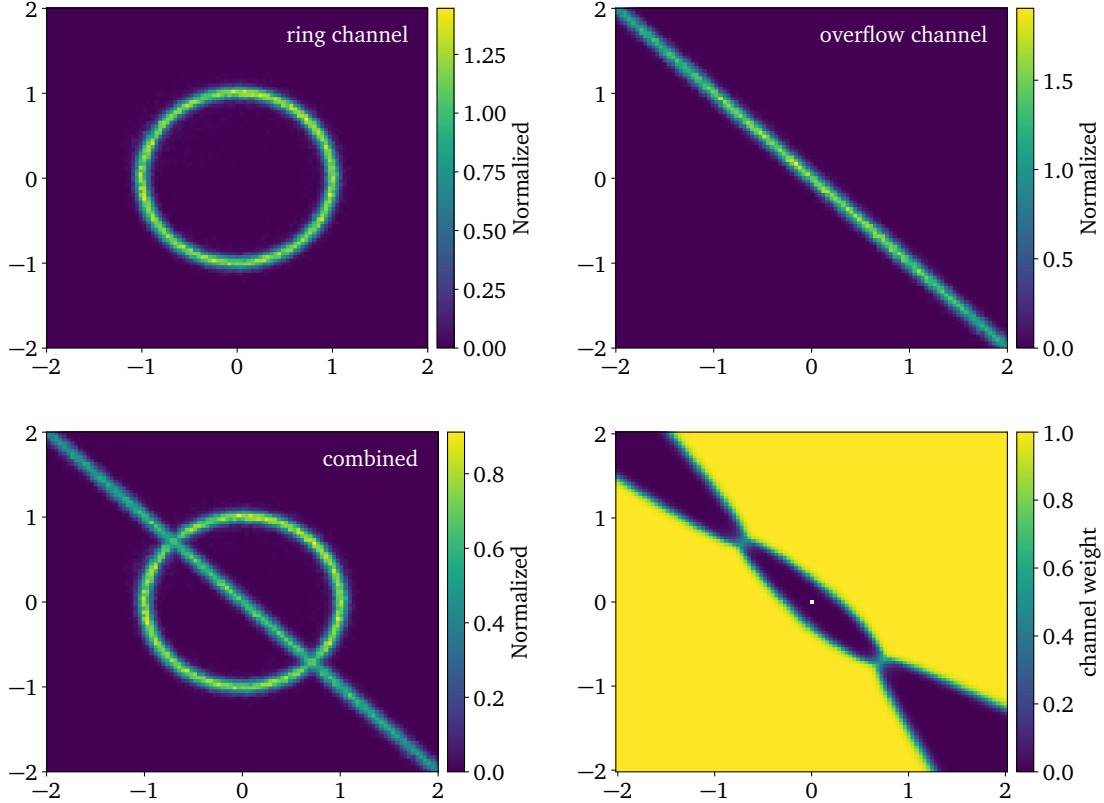

Figure 9: Distribution learned by a 2-channel integrator with a ring mapping and a flat mapping and flat prior. Upper: individual channels, not weighted by channel weights; Lower: combined distribution and channel weight of the ring channel.

**Results**

As a first check of our combined training of the channel weight and channel mapping networks, we use the network setup given in Tab. 2 with one, two or three channels with flat analytic mappings. Because expressive spline-based coupling blocks can learn topologically challenging distributions without a need for multiple channels, we use simple affine coupling blocks. Ideally, our network should automatically define channels removing any topological problems for each individual channel. While this sometimes converges to a reasonable result starting from a flat prior, we found that a mode-specific weight prior led to much more stable results. In detail, we used a prior that encourages one or two channels to focus on the ring and the other to focus on the line. Still, there is a large variation between the results of different trainings. Some examples for the total and channel-wise distributions for different numbers of channels are shown in Fig. 8. The relative uncertainties for different numbers of channels are given in Tab. 2. The performance improves significantly after adding more channels. However, these results are highly sensitive to the choice of the hyperparameters. This suggests that an unsupervised approach to channel partitioning, while theoretically possible, might not be optimal in practice.

Next, we can start with the analytic mapping of the ring and combine it with a flat mapping. Because the ring mapping greatly simplifies the training, we can reduce the number of INN parameters. Results for this combination of learned channels and channel weights is shown in Fig. 9. In the upper panels, we see that the flat channel learns the line without any connection to the pre-defined ring. The integration uncertainties are given in Tab. 2. They show that we

can define overflow channels to extract features that are not captured by pre-defined mappings. The combined distribution in the lower panel closely matches the truth. The channel weights exhibit a clean cut between the two channels with a weight close to 0.5 in the two points where the ring and the line cross. In addition, we show the relative uncertainties for a two-channel integrator with a ring and line mapping and a three-channel integrator with a ring, line and flat mapping in Tab. 2. It can be seen that using a line mapping instead of a flat mapping further improves the performance. Adding an additional flat mapping as an overflow channel is not beneficial since two channels are already sufficient to map out all the features and it just increases the complexity of the training. In all three cases, the relative uncertainties improve compared to the trainings with flat mappings only.

## 5  Drell-Yan plus Z′ at the LHC

After showing how to improve the integration of one- and two-dimensional toy examples, we now use MADNIS for an actual LHC process. To keep things simple, while still challenging all components of our framework, we consider the Drell-Yan process with an additional Z′-resonance,

$$pp \rightarrow \gamma, Z^*, Z'^* \rightarrow e^+ e^-, \tag{50}$$

assuming

$$M_{Z'} = 400.0 \text{ GeV}, \qquad \Gamma_{Z'} = 0.5 \text{ GeV}, \tag{51}$$

for 13 TeV center-of-mass energy. We use the leading-order NNPDF4.0 PDF set [54] with a fixed factorization scale $\mu_F = M_Z$ and $\alpha_s(M_Z) = 0.118$. In the four-flavor scheme we neglect b quarks in the initial state. The Z-parameters are $M_Z = 91.19$ GeV and $\Gamma_Z = 2.44$ GeV. We define the fiducial phase space by requiring only

$$m_{e^+ e^-} > 15 \text{ GeV}. \tag{52}$$

**Implementation details**

To maintain full control, we implement the MADNIS components directly in TENSORFLOW, including the matrix element and the phase-space mappings. The calculation of a hadronic scattering cross section requires many ingredients which need to combined efficiently to achieve precise numerical results. In detail, we implement

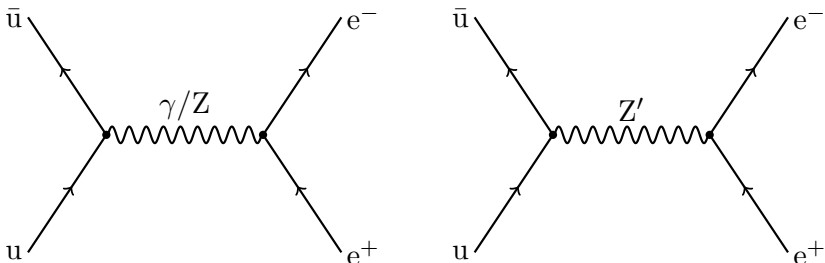

Figure 10: Example LO Feynman diagrams contributing to the Z′-extended Drell-Yan process pp → e⁺e⁻ for one partonic channel.

1. the full squared spin-color averaged/summed LO amplitude

$$\langle |\mathcal{M}|^2 \rangle = \frac{1}{4N_c} \sum_{\text{spins}} |\mathcal{M}_\gamma + \mathcal{M}_Z + \mathcal{M}_{Z'}|^2 , \tag{53}$$

   with $N_c = 3$. As the amplitude is implemented in TENSORFLOW, we can evaluate it in a vectorized form on a CPU and GPU and have access to its gradient, an option we do not use in this study, but plan to use in the future.

2. the hadronic cross section as a convolution of the partonic cross section with the PDFs,

$$\sigma_{pp} = \sum_{a,b} \int_0^1 dx_1 dx_2 \, f_a(x_1) f_b(x_2) \, \hat{\sigma}_{ab}(x_1 x_2 s) . \tag{54}$$

   We use LHAPDF6 [55] and implement our own PYTHON interface to efficiently evaluate large event batches.

3. a multi-channel integration, where we define suitable mappings associated with the different Feynman diagrams.

The hadronic phase space is expressed in terms of $\{x_1, x_2, \cos\theta, \phi\}$. The sampling requires a mapping from the unit hypercube $U = [0,1]^4$ to the two-particle phase space. We implement this mapping sequentially as

$$\begin{aligned}
G_1: &\quad \{y_1, y_2, y_3, y_4\} \rightarrow \{s, y_2, y_3, y_4\}, \\
G_2: &\quad \{s, y_2, y_3, y_4\} \rightarrow \{x_1, x_2, \cos\theta, \phi\},
\end{aligned} \tag{55}$$

where the first step takes into account the propagator structure, so the substitution $y_1 \rightarrow s$ maps out the two mass peaks or the photon propagator. For a resonance with mass $M$ and width $\Gamma$, the standard mapping is again the Breit-Wigner mapping of Eq.(41) [53,56]

$$\begin{aligned}
s(y_1) &= M^2 + M\Gamma \tan\left[\omega_{\min} + (\omega_{\max} - \omega_{\min})y_1\right], \\
g_1(s) &= \frac{1}{\omega_{\max} - \omega_{\min}} \frac{M\Gamma}{(s - M^2)^2 + M^2\Gamma^2},
\end{aligned} \tag{56}$$

where the limits $s = s_{\min} \ldots s_{\max} = 4E_{\text{beam}}^2$ translate into

$$\omega_{\min,\max} = \arctan \frac{s_{\min,\max}^2 - M^2}{M\Gamma} . \tag{57}$$

For the massless photon we instead use the mapping

$$\begin{aligned}
s(y_1) &= \left[ y_1 s_{\max}^{1-\nu} + (1 - y_1) s_{\min}^{1-\nu} \right]^{1/(1-\nu)}, \\
g_1(s) &= \frac{1 - \nu}{s^\nu \left( s_{\max}^{1-\nu} - s_{\min}^{1-\nu} \right)} .
\end{aligned} \tag{58}$$

The hyperparameter $\nu \neq 1$ can be tuned, but we stick to the naive assumption $\nu = 2$. In the second step, we map to $\{x_1, x_2, \cos\theta, \phi\}$ using

$$\begin{aligned}
x_1 &= \left( \frac{s}{s_{\max}} \right)^{y_2}, \qquad x_2 = \left( \frac{s}{s_{\max}} \right)^{1-y_2}, \\
\cos\theta &= 2y_3 - 1, \qquad \phi = 2\pi y_4 - \pi, \qquad \text{with} \quad g_2 = -\frac{s_{\max}}{4\pi \log(x_1 x_2)} .
\end{aligned} \tag{59}$$

We test our numerical setup by computing the fiducial cross section and comparing the result to the standard MG5AMC prediction of $\sigma = (4349.7 \pm 0.32)$ pb to a relative deviation of $10^{-5}$.

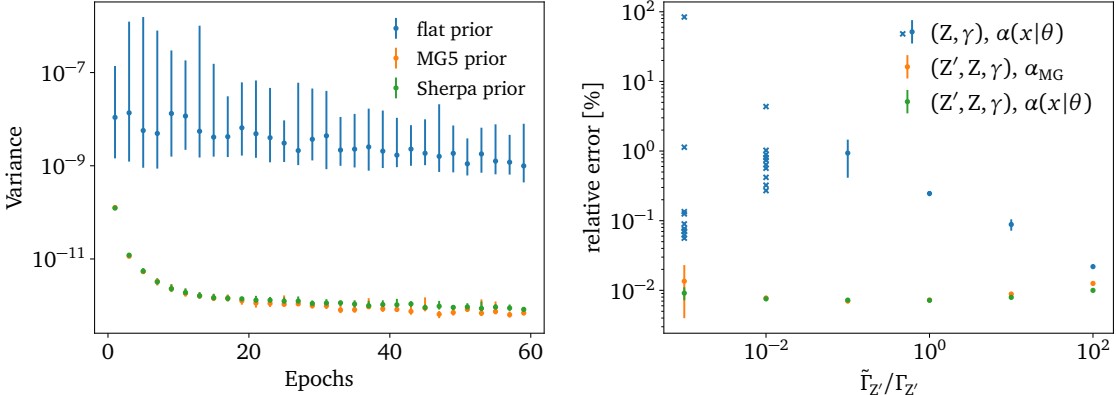

Figure 11: Left: mean and spread (5% to 95% percentile) of 25 evaluations of the variance for three priors of the network weights $\alpha$. Right: integration error as a function of $\Gamma_{Z'}$ for two and three channels, with and without trained channel weights. We give means and standard deviations for ten runs, or the individual results in case of large variation. For very narrow peaks, the two-channel integrator misses the $Z'$ peak entirely.

**Choice of mappings and priors**

While for the simple parametric toy models affine [35,36] coupling blocks were sufficient when combined with a multi-channel strategy, the rich phase-space structure in the $Z'$-extended Drell-Yan process benefits from rational-quadratic spline blocks [57]. Another advantage of spline blocks is that they are naturally defined on a compact domain which makes them especially well-suited for mappings between unit-hypercubes. The other network parameters for this process are given in Tab. 3.

For the toy models we have seen that the choice of mappings and priors is key to a precise integration. This is especially true once we need to cover two narrow peaks in $M_{e^+e^-}$. We confirm this using our network trained with a flat prior, the SHERPA-like prior in Eq.(7), and the MG5AMC-like prior in Eq.(8). After every second epoch, we extract the variance of the integrand from 25 batches of generated samples. The mean and spread of these variances are shown in the left panel of Fig. 11. For both non-flat priors, the variance is stable and converges in the course of the training. In contrast, the flat prior leads to a much larger and unstable variance. Compared to the physics-informed priors the convergence is extremely slow. We follow the standard setup of LHC event generators and include the available physics information through the MG5AMC-like prior of Eq.(8).

Table 3: Hyperparameters of the INN and the channel weight network (CWnet) for the integration of the Drell-Yan + $Z'$ cross section.

| Parameter | Value | | Parameter | Value |
|---|---|---|---|---|
| Loss function | variance | | Coupling blocks | rational-quadratic splines |
| Learning rate | 0.001 | | Permutations | exchange |
| LR schedule | inverse time decay | | Blocks | 6 |
| Decay rate | 0.01 | | Subnet hidden nodes | 16 |
| Batch size | 10000 | | Subnet layers | 2 |
| Epochs | 60 | | CWnet layers | 2 |
| Batches per epoch | 50 | | CWnet hidden nodes | 16 |
| | | | Activation function | leaky ReLU |

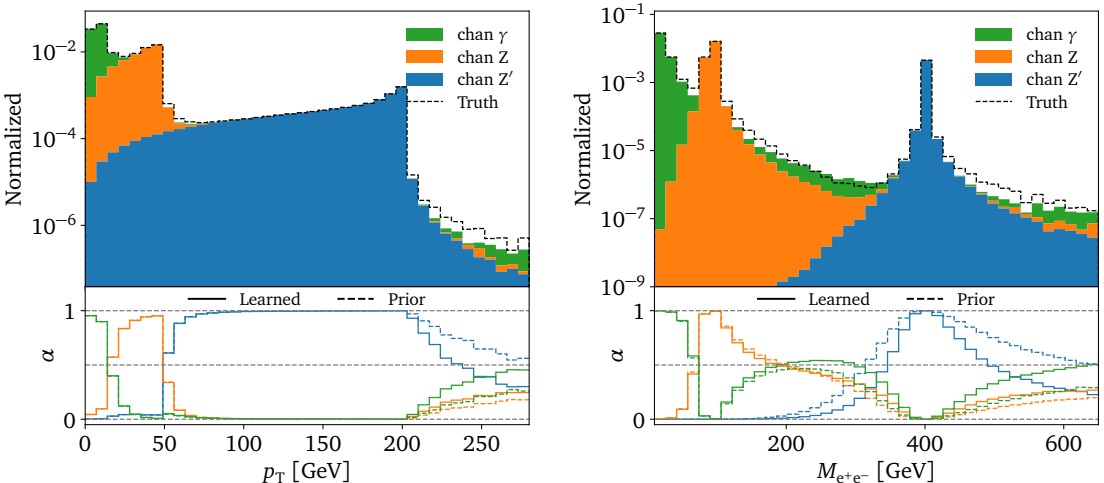

Figure 12: Learned $p_\mathrm{T}$ and $M_{\mathrm{e^+e^-}}$ distributions for the $Z'$-extended Drell-Yan process. In the lower panels we show the learned channel weights.

Second, a powerful physics-informed mapping becomes increasingly important for integrands with narrower features. To this end, we vary the $Z'$-width over several orders of magnitude around the central value given in Eq.(51),

$$\tilde{\Gamma}_{Z'} = \Gamma_{Z'} \times \{10^{-3}, 10^{-2}, 10^{-1}, 10^0, 10^1, 10^2\},\tag{60}$$

while keeping the Z -width constant. In the right panel of Fig. 11, we first compare a two-channel integrator with mappings tailored for the Z and photon diagrams with a three-channel integrator with an additional mapping for the $Z'$. For the three-channel setup, we either fix the channel weights to the MG5AMC prior or train them from this prior. For all three scenarios we give the relative error of the phase-space integral. While the error remains small for the three-channel integrator, even for very narrow decay widths, the integration rapidly degrades for two channels only. For the two narrowest $Z'$-peaks we see a large spread in the variance combined with an overconfident error estimate, indicating that the sampling misses the peak altogether. For the three-channel setup the trainable channel weights lead to a small improvement over the fixed channel weights, mostly for large $\tilde{\Gamma}_{Z'}$. This reflects the fact that for negligible interferences the MG5AMC choice of channel weights is essentially optimal.

In Fig. 12, we look at the phase-space coverage for the distinctive $p_\mathrm{T}$ and $M_{\mathrm{e^+e^-}}$ distributions. We show the learned local channel weights for a three-channel integrator starting from the MG5AMC prior. In agreement with the above result the channel weight network mostly learns small corrections to the prior. Each channel dominates an $M_{\mathrm{e^+e^-}}$ region and the combined distributions are in good agreement with the truth. This means each channel focuses on a single task, as defined by the initialization, rather than learning the full distribution.

**Buffered training**

Even though the integrand for our modified Drell-Yan process is computationally cheap, we can still use it as a test case for our new buffered training. Specifically, we first train the network online for one epoch and save all samples generated during that epoch. Then, we train the network for $k_\mathrm{buff}$ epochs on the saved samples, shuffling them every time. After that, we discard the saved samples. We find that this training schedule works well for our application, but it can be easily adapted for other application. For example, we can save samples from more than one online training epoch.

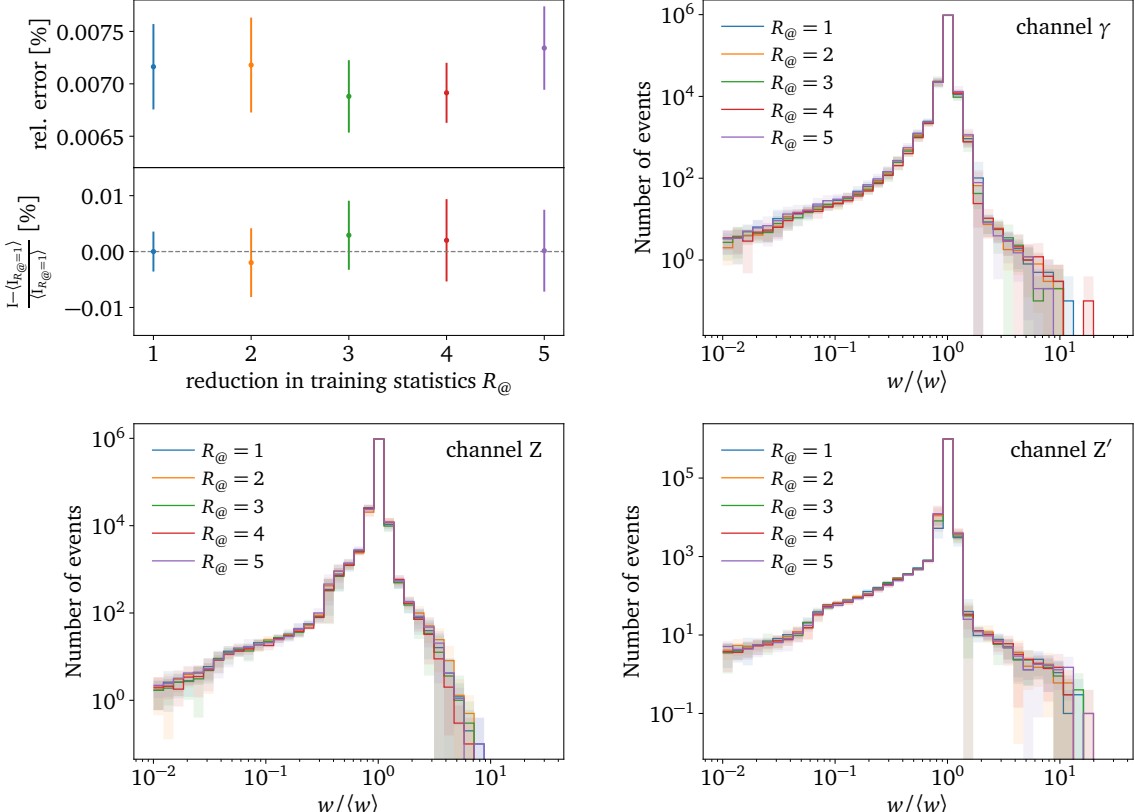

Figure 13: Relative integration error (from $10^6$ events), relative deviation from the mean $R_@ = 1$ result, and weight distributions for different reduction factors $R_@$ in training statistics for the Z'-extended Drell-Yan process. The points/lines and error bars/bands show means and standard deviations over ten runs.

To benchmark the buffered training, we continue to train the network for 60 epochs, but replace some of the online training epochs with training on samples following the above schedule. The training cycle is then repeated $60/(k_{\text{buff}} + 1)$ times, and the relative reduction in the training statistics defined in Eq.(28) is

$$R_@ = k_{\text{buff}} + 1, \qquad \text{with} \qquad k_{\text{buff}} = 0, 1, 2, 3, 4. \qquad (61)$$

For each value of $k_{\text{buff}}$ we run our integrator ten times. The relative integration error, the relative deviation from the mean $R_@ = 1$ result, and the weight distributions for the three different channels are shown in Fig. 13. Even for a reduction of the training statistics by a factor five the performance of the integrator — in terms of the relative error and the weight distribution — matches the pure online training. Even in this simple case, where the evaluation time for the integrand is negligible, the training time can be reduced by around 20% because of the lower number of INN evaluations.

As a side remark, we have tested how different choices for the permutation layer affect the integration. While the trainable soft permutations perform much better than the fixed soft permutations, soft permutations perform slightly worse than simple exchange permutations for this low-dimensional problem. The reason for this is that the features the flow has to learn are almost perfectly aligned with the axis of the chosen parametrization without any rotation, and that spline blocks require us to nest the soft permutations between logit and sigmoid functions, which leads to potentially slower convergence.

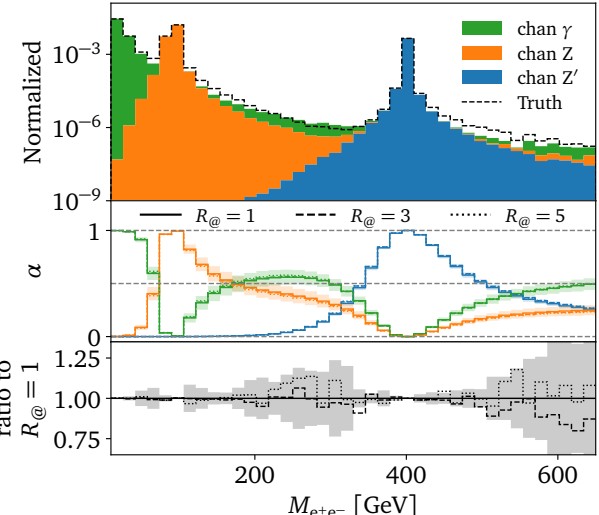

Figure 14: Learned $M_{e^+e^-}$ distributions for the $Z'$-extended Drell-Yan process. The upper panel is the same as in Fig. 12, the middle panel shows the learned channel weights, and the lower panel shows the ratio of the combined distribution to pure online training for reduction factors $R_@$ in training statistics, see Eq.(61). The lines in the lower two panels are obtained by averaging over ten independent trainings. The error envelopes are only shown for $R_@ = 1$.

# 6 Outlook

We introduced the new, comprehensive MADNIS approach to importance sampling and multi-channel integration. The bijective variable transformations behind importance sampling suggest using normalizing flows, in our case an INN which is equally fast in both directions. For LHC event generators, this ML-integrator needs to be embedded in a common framework with multi-channel integration. We have shown how to efficiently combine normalizing flows with a multi-channel strategy by defining local and trainable multi-channel weights. Finally, we developed trainable rotations as a general permutation layer between the INN coupling blocks. They will become beneficial for high-dimensional phase spaces.

For simple parametric examples, we have seen that it is possible to learn optimal channel weights, including a combination with normalizing flows. Moreover, we have shown that it is possible to define single or multiple overflow channels and leave it to the networks to split the complicated topological structure into easy-to-learn substructures. More realistically, we have shown that our framework works for the $Z'$-extended Drell-Yan process, which includes many challenges of a generic LHC process while still having a low-dimensional phase space. In particular, it requires a combination of the normalizing flow with a physics-informed mapping to achieve a precise integration at low computational cost.

A bottleneck for current LHC predictions is increasingly expensive evaluations of the matrix element. To alleviate this problem, we combine expensive online training with buffered sample training. In Fig. 14, we illustrate the performance of the MADNIS methodology, including an effective reduction in training statistics by using buffered training in addition to the standard online training. For our LHC example, our new training scheme can reduce the number of calls to the matrix element by a factor of five without losing precision in the integration.

# Acknowledgements

**Funding information** The authors would like to express special thanks to the Mainz Institute for Theoretical Physics (MITP) of the Cluster of Excellence PRISMA+ (Project ID 39083149), for its hospitality and support. OM, FM and RW acknowledge support by FRS-FNRS (Belgian National Scientific Research Fund) IISN projects 4.4503.16. CK was supported by DOE grant DOE-SC0010008. AB, CK, and TP would like to thank the Baden-Württemberg-Stiftung for funding through the program *Internationale Spitzenforschung*, project *Uncertainties — Teaching AI its Limits* (BWST_IF2020-010). AB would like to acknowledge support by the BMBF for the AI junior group 01IS22079. TH is supported by the DFG Research Training Group GK-1940, *Particle Physics Beyond the Standard Model*. TH's contribution to this project was made possible by funding from the Carl-Zeiss-Stiftung. Computational resources have been provided by the supercomputing facilities of the Université catholique de Louvain (CISM/UCL) and the Consortium des Équipements de Calcul Intensif en Fédération Wallonie Bruxelles (CÉCI) funded by the Fond de la Recherche Scientifique de Belgique (F.R.S.-FNRS) under convention 2.5020.11 and by the Walloon Region. The work of JI was supported by the Fermi National Accelerator Laboratory (Fermilab), a U.S. Department of Energy, Office of Science, HEP User Facility. Fermilab is managed by Fermi Research Alliance, LLC (FRA), acting under Contract No. DE–AC02–07CH11359. This project was supported by the Deutsche Forschungsgemeinschaft (DFG, German Research Foundation) under grant 396021762 – TRR 257 *Particle Physics Phenomenology after the Higgs Discovery* and through Germany's Excellence Strategy EXC 2181/1 - 390900948 *The Heidelberg STRUCTURES Excellence Cluster*.

# A  Buffered losses and training

Splitting the integral of Eq.(1) using the trained weights defined in Eq.(18) we first define normalized channel-wise probability distributions as

$$
I[f] = \sum_i I_i(\theta) = \sum_i \int_\Phi \mathrm{d}^d x \, \alpha_i(x|\theta) f(x)
$$
$$
\implies \quad p_i(x|\theta) = \frac{\alpha_i(x|\theta) f(x)}{I_i(\theta)}, \tag{A.1}
$$

where the channel-wise integrals change during training. The goal is to approximate these probability distributions with a network function in terms of the weights $\varphi$,

$$
p_i(x|\theta) \approx g_i(x|\varphi). \tag{A.2}
$$

The implicit dependence of $g_i(x|\varphi)$ on $\theta$ enters through this training objective. To quantify the agreement between the two functions we can use a range of divergences $D$, all summed over the channels,

$$
\mathcal{L} = \sum_i a_i \, D_i[p_i; g_i], \tag{A.3}
$$

with arbitrary weights $a_i$. For a combined training of the channel weights ($\theta$) and the importance sampling ($\varphi$) we have to be careful when updating the losses based on these divergences.

## Neyman $\chi_N^2$ divergence

The first divergence we can use to define our loss is the Neyman-$\chi_N^2$ divergence

$$
\begin{aligned}
D_{\chi_N^2,i} &= \int_\Phi d^d x \, \frac{[p_i(x|\theta) - g_i(x|\varphi)]^2}{g_i(x|\varphi)} \\
&= \int_\Phi d^d x \, \frac{p_i(x|\theta)^2}{g_i(x|\varphi)} - 2 \underbrace{\int_\Phi d^d x \, p_i(x|\theta)}_{=1} + \underbrace{\int_\Phi d^d x \, g_i(x|\varphi)}_{=1} .
\end{aligned}
\tag{A.4}
$$

To minimize $D_{\chi^2,i}$, we need its gradient with respect to $\varphi$ and $\theta$.

$$
\begin{aligned}
\nabla_\varphi D_{\chi_N^2,i} &= \int_\Phi d^d x \, p_i(x|\theta)^2 \, \nabla_\varphi \frac{1}{g_i(x|\varphi)} = -\int_\Phi d^d x \, \frac{p_i(x|\theta)^2}{g_i(x|\varphi)} \, \nabla_\varphi \log g_i(x|\varphi) \\
&= \left\langle -\frac{p_i(x|\theta)^2}{q_i(x|\hat\varphi) g_i(x|\varphi)} \, \nabla_\varphi \log g_i(x|\varphi) \right\rangle_{x \sim q_i(x|\hat\varphi)} , \\
\nabla_\theta D_{\chi_N^2,i} &= 2 \int_\Phi d^d x \, \frac{p_i(x|\theta)}{g_i(x|\varphi)} \, \nabla_\theta p_i(x|\theta) = 2 \int_\Phi d^d x \, \frac{p_i(x|\theta)^2}{g_i(x|\varphi)} \, \nabla_\theta \log p_i(x|\theta) \\
&= 2 \left\langle \frac{p_i(x|\theta)^2}{q_i(x|\hat\varphi) g_i(x|\varphi)} \, \nabla_\theta \log p_i(x|\theta) \right\rangle_{x \sim q_i(x|\hat\varphi)} .
\end{aligned}
\tag{A.5}
$$

Note that we evaluate the integrals by sampling from a proposal function $x \sim q_i(x|\hat\varphi)$, which can be either a totally independent function that is easy to sample from and the dependence of $\varphi$ drops out, i.e. $q_i(x|\hat\varphi) = q_i(x)$, or it is directly linked to the importance weight $q_i(x|\hat\varphi) = g_i(x|\hat\varphi)$ possibly depending on different network weights $\hat\varphi \neq \varphi$ which is relevant for the buffered training as described in Sec. 3.3. The loss functions are then given by

$$
\begin{aligned}
\mathcal{L}_{\chi_N^2}^{\text{int}} &= -\sum_i a_i \left\langle \frac{p_i(x|\theta)^2}{q_i(x|\hat\varphi) g_i(x|\varphi)} \, \log g_i(x|\varphi) \right\rangle_{x \sim q_i(x|\hat\varphi)} , \\
\mathcal{L}_{\chi_N^2}^{\text{weights}} &= 2\sum_i a_i \left\langle \frac{p_i(x|\theta)^2}{q_i(x|\hat\varphi) g_i(x|\varphi)} \, \log p_i(x|\theta) \right\rangle_{x \sim q_i(x|\hat\varphi)} ,
\end{aligned}
\tag{A.6}
$$

where the red expressions have to be evaluated without gradient calculation. Note that $p_i(x|\theta)$ indirectly also depends on $\hat\varphi$ as the samples are drawn from $x \sim q_i(x|\hat\varphi)$. However, we do not need a gradient calculation for $p_i$.

## Variance loss

Alternatively, we can minimize the variance of the normalized functions $p_i(x|\theta)/g_i(x|\varphi)$,

$$
\begin{aligned}
\mathbb{V}_i &= \left\langle \frac{p_i(x|\theta)^2}{g_i(x|\varphi)^2} \right\rangle_{x \sim g_i(x|\varphi)} - \left\langle \frac{p_i(x|\theta)}{g_i(x|\varphi)} \right\rangle_{x \sim g_i(x|\varphi)}^2 \\
&= \int_\Phi d^d x \, \frac{p_i(x|\theta)^2}{g_i(x|\varphi)} - \underbrace{\left( \int_\Phi d^d x \, p_i(x) \right)^2}_{=1} .
\end{aligned}
\tag{A.7}
$$

This is the same expression as $D_{\chi_N^2}$ in Eq.(A.4), so the losses are given by Eq.(A.6). Note that we can write Eq.(A.7) into a MC estimate using the sampling $x \sim q_i(x|\hat\varphi)$,

$$
\mathbb{V}_i = \left\langle \frac{p_i(x|\theta)^2}{g_i(x|\varphi) q_i(x|\hat\varphi)} \right\rangle_{x \sim q_i(x|\hat\varphi)} - \left\langle \frac{p_i(x|\theta)}{q_i(x|\hat\varphi)} \right\rangle_{x \sim q_i(x|\hat\varphi)}^2 .
\tag{A.8}
$$

## Pearson $\chi_P^2$ divergence

A similar choice is the Pearson-$\chi_P^2$ divergence,

$$
\begin{aligned}
D_{\chi_P^2, i} &= \int_\Phi d^d x \, \frac{(g_i(x|\varphi) - p_i(x|\theta))^2}{p_i(x|\theta)} \\
&= \int_\Phi d^d x \, \frac{g_i(x|\varphi)^2}{p_i(x|\theta)} - 2 \underbrace{\int_\Phi d^d x \, g_i(x|\varphi)}_{=1} + \underbrace{\int_\Phi d^d x \, p_i(x|\theta)}_{=1} \, .
\end{aligned}
\tag{A.9}
$$

To minimize $D_{\chi_P^2, i}$ we need the two gradients

$$
\begin{aligned}
\nabla_\varphi D_{\chi_P^2, i} &= 2 \int_\Phi d^d x \, \frac{g_i(x|\varphi)}{p_i(x|\theta)} \nabla_\varphi g_i(x|\varphi) = 2 \int_\Phi d^d x \, \frac{g_i(x|\varphi)^2}{p_i(x|\theta)} \nabla_\varphi \log g_i(x|\varphi) \\
&= 2 \left\langle \left( \frac{g_i(x|\varphi)^2}{p_i(x|\theta) q_i(x|\hat\varphi)} \right) \nabla_\varphi \log g_i(x|\varphi) \right\rangle_{x \sim q_i(x|\hat\varphi)}, \\
\nabla_\theta D_{\chi_P^2, i} &= \int_\Phi d^d x \, g_i(x|\varphi)^2 \nabla_\theta \frac{1}{p_i(x|\theta)} = -\int_\Phi d^d x \, \frac{g_i(x|\varphi)^2}{p_i(x|\theta)} \nabla_\theta \log p_i(x|\theta) \\
&= -\left\langle \left( \frac{g_i(x|\varphi)^2}{p_i(x|\theta) q_i(x|\hat\varphi)} \right) \nabla_\varphi \log p_i(x|\theta) \right\rangle_{x \sim q_i(x|\hat\varphi)}.
\end{aligned}
\tag{A.10}
$$

The corresponding losses can be written as

$$
\begin{aligned}
\mathcal{L}_{\chi_P^2}^{\text{int}} &= 2 \sum_i a_i \left\langle \left( \frac{g_i(x|\varphi)^2}{p_i(x|\theta) q_i(x|\hat\varphi)} \right) \log q_i(x|\varphi) \right\rangle_{x \sim q_i(x|\hat\varphi)}, \\
\mathcal{L}_{\chi_P^2}^{\text{weights}} &= -\sum_i a_i \left\langle \left( \frac{g_i(x|\varphi)^2}{p_i(x|\theta) q_i(x|\hat\varphi)} \right) \log p_i(x|\theta) \right\rangle_{x \sim q_i(x|\hat\varphi)},
\end{aligned}
\tag{A.11}
$$

where, again, the red expressions have to be evaluated without gradient calculation.

## KL-divergence

As a fourth option, we can use the KL-divergence to train the network,

$$
\begin{aligned}
D_{\text{KL}, i} &= \int_\Phi d^d x \, p_i(x|\theta) \log \frac{p_i(x|\theta)}{g_i(x|\varphi)} \\
&= \int_\Phi d^d x \, p_i(x|\theta) \log p_i(x|\theta) - \int_\Phi d^d x \, p_i(x|\theta) \log g_i(x|\varphi).
\end{aligned}
\tag{A.12}
$$

To minimize $D_{\text{KL}, i}$ with respect to $\varphi$ we only need to consider the second term, which is the cross entropy,

$$
\nabla_\varphi D_{\text{KL}, i} = -\int_\Phi d^d x \, p_i(x|\theta) \nabla_\varphi \log q_i(x|\varphi) = -\left\langle \frac{p_i(x|\theta)}{q_i(x|\hat\varphi)} \nabla_\varphi \log g_i(x|\varphi) \right\rangle_{x \sim q_i(x|\hat\varphi)}.
\tag{A.13}
$$

To train the channel weight we evaluate

$$
\begin{aligned}
\nabla_\theta D_{\text{KL},i} &= \int_\Phi \mathrm{d}^d x \, \nabla_\theta p_i(x|\theta) \, \log p_i(x|\theta) + \int_\Phi \mathrm{d}^d x \, p_i(x|\theta) \, \nabla_\theta \log p_i(x|\theta) \\
&\quad - \int_\Phi \mathrm{d}^d x \, \nabla_\theta p_i(x|\theta) \log g_i(x|\varphi) \\
&= \int_\Phi \mathrm{d}^d x \, p_i(x|\theta) \Big( 1 + \log \frac{p_i(x|\theta)}{g_i(x|\varphi)} \Big) \nabla_\theta \log p_i(x|\theta) \\
&= \Big\langle \frac{p_i(x|\theta)}{q_i(x|\hat\varphi)} \Big( 1 + \log \frac{p_i(x|\theta)}{g_i(x|\varphi)} \Big) \nabla_\theta \log p_i(x|\theta) \Big\rangle_{x \sim q_i(x|\hat\varphi)} .
\end{aligned}
\tag{A.14}
$$

The two loss functions are then

$$
\begin{aligned}
\mathcal{L}_{\text{KL}}^{\text{int}} &= -\sum_i a_i \Big\langle \frac{p_i(x|\theta)}{q_i(x|\hat\varphi)} \log g_i(x|\varphi) \Big\rangle_{x \sim q_i(x|\hat\varphi)} , \\
\mathcal{L}_{\text{KL}}^{\text{weights}} &= \sum_i a_i \Big\langle \frac{p_i(x|\theta)}{q_i(x|\hat\varphi)} \Big( 1 + \log \frac{p_i(x|\theta)}{g_i(x|\varphi)} \Big) \log p_i(x|\theta) \Big\rangle_{x \sim q_i(x|\hat\varphi)} .
\end{aligned}
\tag{A.15}
$$

Comparing the first loss to Eq.(A.6), we see that the log-likelihood is only weighted with a single MC weight, so the $\chi_N^2$ loss penalizes large discrepancies stronger, specifically, low values of $q_i$ in regions of high density $p_i$.

**Reverse KL-divergence**

The mode-seeking behavior of the reverse KL-divergence [58]

$$
D_{\text{RKL},i} = \int_\Phi \mathrm{d}^d x \, g_i(x|\varphi) \log \frac{g_i(x|\varphi)}{p_i(x|\theta)} ,
\tag{A.16}
$$

can be beneficial in the training of the normalizing flow, as it pushes the flow to assign zero density where $p_i(x|\theta)$ is zero and focuses on the modes of $p_i(x|\theta)$. Unlike for the forward KL-divergence, the gradient with respect to $\varphi$ is now more complex,

$$
\begin{aligned}
\nabla_\varphi D_{\text{RKL},i} &= \int_\Phi \mathrm{d}^d x \, \nabla_\varphi g_i(x|\varphi) \log g_i(x|\varphi) + \int_\Phi \mathrm{d}^d x \, g_i(x|\varphi) \nabla_\varphi \log g_i(x|\varphi) \\
&\quad - \int_\Phi \mathrm{d}^d x \, \nabla_\varphi g_i(x|\varphi) \log p_i(x|\theta) \\
&= \int_\Phi \mathrm{d}^d x \, g_i(x|\varphi) \Big( 1 + \log \frac{g_i(x|\varphi)}{p_i(x|\theta)} \Big) \nabla_\varphi \log g_i(x|\varphi) \\
&= \Big\langle \frac{g_i(x|\varphi)}{q_i(x|\hat\varphi)} \Big( 1 + \log \frac{g_i(x|\varphi)}{p_i(x|\theta)} \Big) \nabla_\varphi \log g_i(x|\varphi) \Big\rangle_{x \sim q_i(x|\hat\varphi)} ,
\end{aligned}
\tag{A.17}
$$

while the training of the channel weights just requires

$$
\nabla_\theta D_{\text{RKL},i} = -\nabla_\theta \int_\Phi \mathrm{d}^d x \, g_i(x|\varphi) \log p_i(x|\theta) = -\Big\langle \frac{g_i(x|\varphi)}{q_i(x|\hat\varphi)} \nabla_\theta \log p_i(x|\theta) \Big\rangle_{x \sim q_i(x|\hat\varphi)} .
\tag{A.18}
$$

Consequently, we can write for the total loss function

$$
\begin{aligned}
\mathcal{L}_{\text{RKL}}^{\text{int}} &= \sum_i a_i \Big\langle \frac{g_i(x|\varphi)}{q_i(x|\hat\varphi)} \Big( 1 + \log \frac{g_i(x|\varphi)}{p_i(x|\theta)} \Big) \log g_i(x|\varphi) \Big\rangle_{x \sim q_i(x|\hat\varphi)} , \\
\mathcal{L}_{\text{RKL}}^{\text{weights}} &= -\sum_i a_i \Big\langle \frac{g_i(x|\varphi)}{q_i(x|\hat\varphi)} \log p_i(x|\theta) \Big\rangle_{x \sim q_i(x|\hat\varphi)} .
\end{aligned}
\tag{A.19}
$$

In contrast to the KL divergence, which is mass-distributing, this loss for the normalizing flow only includes the logarithm of the MC weight and an additional positive $\log g_i(x|\varphi)$ term, reflecting the mode-seeking behavior. Furthermore, the RKL loss only requires $p_i(x|\theta)$ and not its derivative $\nabla_x p_i(x|\theta)$, when taking the gradients before reparametrization.

**Single-pass gradient computation**

Finally, when evaluating one of the above-described losses during online training, we need to be careful. In practice, we want to follow the steps

1. sample points $y \sim$ uniform;

2. map $y \to x_\varphi \equiv x(y|\varphi) = \overline{G}(y|\varphi)$ and evaluate the density $\bar{g}(y|\varphi) = g(x_\varphi|\varphi)^{-1}$;

3. evaluate the target function $f(x_\varphi) \sim p(x_\varphi)$;

4. calculate a divergence-based loss between $p(x_\varphi)$ and $g(x_\varphi|\varphi)$;

5. compute gradients of the loss and optimize the network.

The training workflow is summarized in Fig. 15 and also shows the backpropagation of the gradients coming from the loss function. For example, the KL-loss is

$$D_{\text{KL}}[p(x_\varphi), g(x_\varphi|\varphi)] = \int \mathrm{d}^d y \; p(x_\varphi) \log \frac{p(x_\varphi)}{g(x_\varphi|\varphi)}\Bigg|_{x_\varphi = \overline{G}(y|\varphi)} . \tag{A.20}$$

For optimization during training, we require its gradient with respect to the network parameters $\varphi$

$$\nabla_\varphi D_{\text{KL}}[p(x_\varphi), g(x_\varphi|\varphi)], \tag{A.21}$$

which would also require us to calculate

$$\frac{\partial p(x_\varphi)}{\partial \varphi} = \frac{\partial p(x_\varphi)}{\partial x_\varphi} \frac{\partial x_\varphi}{\partial \varphi} . \tag{A.22}$$

However, the first term is intractable for common event generators, as the amplitude is not differentiable. To circumvent this limitation, we define the loss as a proper function of $x$, such that we do not require the gradient of $p(x)$. This means, we replace Eq.(A.20) with

$$D_{\text{KL}}[p(x), g(x|\varphi)] = \int \mathrm{d}^d x \, p(x) \log \frac{p(x)}{g(x|\varphi)} . \tag{A.23}$$

In this form, we also need the density $g(x|\varphi)$ as a proper function of $x$ to obtain the correct gradients. We illustrate this for a two-dimensional toy flow $G$ with one trainable parameter $\varphi$,

$$\begin{aligned} \text{forward } y = G(x|\varphi): \quad & y_1 = x_1 + \varphi, \quad & y_2 = x_2 \cdot \exp x_1, \\ \text{inverse } x = \overline{G}(y|\varphi): \quad & x_1 = y_1 - \varphi, \quad & x_2 = y_2 \cdot \exp[-y_1 + \varphi]. \end{aligned} \tag{A.24}$$

The corresponding Jacobians are

$$g(x|\varphi) = \left|\frac{\partial G(x|\varphi)}{\partial x}\right| = \exp x_1, \quad \text{and} \quad \bar{g}(y|\varphi) = \left|\frac{\partial \overline{G}(y|\varphi)}{\partial y}\right| = \exp[-y_1 + \varphi]. \tag{A.25}$$

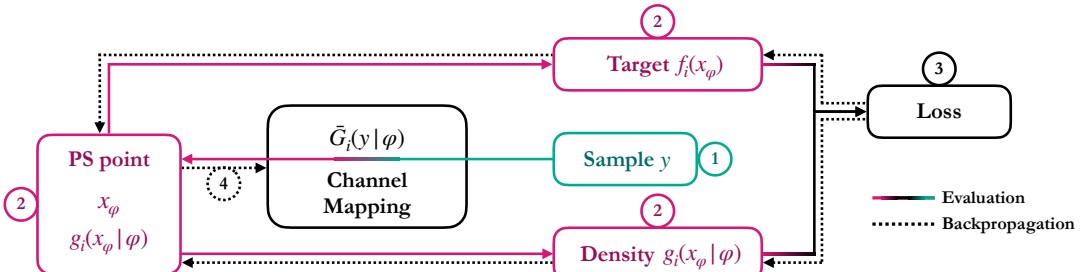

Figure 15: Workflow of the single-pass training of the INN.

While $g(x|\varphi) = \bar{g}(y|\varphi)^{-1}$, $g$ is still a function of $x$ and $\bar{g}$ is a function of $y$. Their gradients with respect to $\varphi$ will therefore be different,

$$\frac{\partial g(x|\varphi)}{\partial \varphi} = 0, \qquad \text{and} \qquad \frac{\partial \bar{g}(y|\varphi)}{\partial \varphi} = \exp[-y_1 + \varphi] = \exp[-x_1(y)]. \tag{A.26}$$

This means that after the inverse pass $x = \overline{G}(y|\varphi)$, which has to be evaluated without gradients to avoid unwanted gradients for $p(x)$, we perform an additional forward pass $y = G(x|\varphi)$, see Figs. 2 and 3. This forward pass evaluates the Jacobian $g(x|\varphi)$ as a proper function of $x$. In contrast, the inverse pass would return the Jacobian $\bar{g}(y|\varphi)$ as a function of $y$ and yield wrong gradients.

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
