# Peer review of "MadNIS -- Neural Multi-Channel Importance Sampling"

_SciPost Physics, doi:SciPost Phys. 15, 141 (2023)_

## Round 1 · Author Response

We are grateful to the referees for their time and thoughtful feedback. We have integrated several suggestions into the text and fixed small typos. Below we give a detailed overview of all changes, including our responses to the referees' comments and questions:

Referee 3:

  1. We added the definition of the unit hypercube where it appears first on page 3.

  2. We have added these references and now cite them in the text accordingly.

  3. As the raw network output is unconstrained before normalization, the right-hand side of Eq.(19) allows for negative channel weights. While this is mathematically allowed and fulfills all necessary requirements for the channel splittings, the weights indeed lose their interpretation as probabilities. We also found that this normalization is numerically unstable and hence used the presented normalization on the left-hand side of Eq.(19) which includes a softmax activation function. We added two more lines to the text to further clarify this.

  4. There was indeed a typo in Eq. (23) and we fixed this in the new version.

  5. As mentioned in the text, the inclusion of channels with zero events during training causes errors in the optimization. To be precise, as each channel is covered by its own normalizing flow, each flow expects an input-tensor with a length > 0 to be able to calculate gradients. If, however, no events are sampled into a certain channel, the input-tensor to the corresponding normalizing flow is None and hence causes a runtime error. We could ignore this channel if no events are sampled into it, however, this might cause that this channel will never be populated again even though this might have only been a fluctuation during optimization. Hence, we decided to make sure each channel is always populated at least by some fraction of events to prevent instabilities. In contrast, after optimization, where we expect these fluctuations to be averaged away, we can safely ignore channels that are not populated during integration and hence no runtime-error occurs.

  6. The rotations we implemented are proper rotations in the R^n. Since they are described by elements of SO(n), they are not distorting features (the transformations are angle preserving) on this space and they are defined for any angle, not necessarily close to the identity. Note, that these rotations are only properly defined on R^n and not on the unit-hypercube, where the rotations would be ill-defined in general. In order to combine these soft permutation with a flow acting on a compact space, let’s assume for instance [0,1]^n, these features have to be mapped onto the full R^n before applying. For instance, a possible combination would be a Logit-SoftPermutation-Sigmoid transformation. However, in this case, rotations which are not multiplicities of Pi/2 will indeed cause distortion near the boundaries of the unit hypercube.

  7. This is indeed an idea we consider in a follow-up project where we try to consider possible symmetries between different channels which could potentially decrease the computational overhead.

  8. As each channel is handled individually and governs its own phase-space mapping as well as its own normalizing flow, we do not expect severe problems in scenarios where phase-space cuts affect different channels in a different way.

  9. At the time being, the paper was not meant to serve as benchmarking of standard MG5aMC versus a ML supplemented scenario but meant to introduce new concepts which are relevant to improve phase-space integration and which has not been covered in previous applications. A proper and more detailed comparison between MadNIS and standard MadGraph is subject to a current follow-up project.

  10. This is indeed a typo. We have fixed this in the new version.

Referee 2:

  1. Conceptually, this is similar to two channel-mappings when in fact three channel mappings would be required to cover all features. This is a scenario we have tested and is illustrated in the right panel of Fig. 11. Indeed, interferences usually also come with a negative effect which might alter the total integrand differently. However, we do not see any limitations to our approach in this case. In addition, the introduction of the overflow channel captures all structures that have not been learned properly by the other channels, as shown in the ring example. We are currently working on a madgraph-specific follow-up in which we will further investigate this and many other questions on several standard processes. If there are any other concerns or ideas the referee thinks would be interesting to look at, please let us know and we can investigate this further in the follow-up paper. We would also like to stretch again that this manuscript here is meant to be conceptual and serves as a proof-of-concept for several new ideas which go way beyond what has been done before in the literature. Hence, we think the manuscript as it is fully serving this purpose and does not need any more additional examples being tested to illustrate our advances.

---

## Round 1 · List of Changes

Except for the changes already mentioned above we further updated and fixed the following:

1. There was a typo in the definition of stratified sampling in Eq.(17) which has been fixed now.

2. There was a smalle error in the ring mapping both in Eq.(48) as well as in the code. We fixed this and rerun
our code with the corrected mapping. We updated the right-hand side of Table 2 and Figure 9 accordingly.

---

## Editorial Decision

published